

# Determination of appropriate land use/cover pattern based on the hydroclimatic regime to support regional ecological management in the agro-pastoral ecotone of northwest China

Yuzuo Zhu[1], Xuefeng Xu[1]

[1] Key Laboratory of West China's Environmental System (Ministry of Education), College of Earth and Environmental Sciences, Lanzhou University, Lanzhou, Gansu 730000, China

*Correspondence to*: Yuzuo Zhu (zhuyz16@lzu.edu.cn)

**Abstract.** The agro-pastoral ecotone of northwest China (APENC) has been experiencing large-scale land use/cover change (LUCC) since 1999 as vegetation restoration projects were implemented. Negative environmental effects of excessive re-vegetation have emerged. However, it remains unclear what the optimal mixture of land use/cover in vegetation restoration is to maintain a sustainable ecohydrological environment in the APENC. In this study, we investigate the different scenarios associated with vegetation restoration in the APENC to examine the hydroclimatic impacts of vegetation restoration and identify the proper land use/cover pattern based on hydroclimatic thresholds (cooling surface and higher water conservation) using the Community Land Model version 5.0 (CLM5.0). Results show that the two main types of LUCC in the study region from 2000 to 2015 were the conversion of bare land to grasslands (BL to GRS) and croplands to grasslands (CL to GRS). The BL to GRS decreased the annual mean temperature by -0.17 ℃, while CL to GRS increased the annual mean temperature by 0.96 ℃; ET changes were 53.32 and -184.42 mm yr$^{-1}$, respectively, leading to an annual spatially averaged land surface temperature (LST) by a cooling range of -0.06 ± 0.15 ℃ and evapotranspiration (ET) increased by a range of 9.70 ± 19.04 mm yr$^{-1}$ in the study region. The correlation coefficients between biogeophysical characteristics and hydroclimatic change indicated surface albedo was the most sensitive surface characteristic in influencing LST and ET in summer and winter for BL to GRS and CL to GRS, while the LAI + SAI also presented the most significant correlation for CL to GRS throughout the year. Additionally, analysis of change in land use/cover pattern from 2000 to 2015 found that some grids experienced drying and warming as re-vegetation projects due to the offsetting effects of two types of LUCC. Our findings suggest the percentage of grasslands, bare land and croplands in the APENC for 2035 approximately is 60 %, 23 % and 11 %, respectively, which will mitigate the drying and warming surface environment in the semi-arid region. The findings provide important information to support long-term regional sustainable development in the APENC and similar regions.

**Keywords**: Land use/cover change, Water conservation, CLM5.0, Land use pattern, Agricultural-pastoral ecotone in Northwest



## 1 Introduction

Land use/cover change (LUCC), such as deforestation, afforestation, grassland restoration and agricultural expansion, affects the interaction of energy and vapor at the interface between land and atmosphere via modifying biogeophysical characteristics, thereby violently modulating climate and hydrology on regional and global scales (Alkama and Cescatti, 2015; Chen and Dirmeyer, 2016; Chen and Dirmeyer, 2017; Davin et al., 2020; Duveiller et al., 2018; Liu et al., 2016; Woodward et al., 2014).

The LUCC has been recognized as one of the key climatic and hydrological mitigation strategies available to governments, especially under global warming and water resources shortages (Arora and Montenegro, 2011; Davin et al., 2014; Findell et al., 2017; Poniatowski et al., 2020). Therefore, examining the impacts of LUCC and then developing optimal land use/cover patterns are crucial for supporting long-term sustainable land management and ecosystem services (Jia et al., 2017a; Zhang et al., 2018).

Generally, statistical analyses based on in situ observations, satellite products and simulated scenarios by numerical models have been adopted widely (Lee et al., 2011; Nkhoma et al., 2021). However, in situ observations are sparsely and unevenly distributed in many regions, and it is difficult to derive data for larger areas due to the constraints of equipment and resources (Li et al., 2021; Zhang et al., 2021). Satellite products with relatively high spatial resolution hardly provide accurate continuous long-term data owing to the satellite just obtaining instantaneous images and uncertainty in the processing methods (Srivastava

et al., 2015; Zhang et al., 2010). While, with significant improvements in numerical models, numerical models have provided an opportunity to not only study multiple variables in high spatial resolution over longer periods but also access to fluxes cycle with one consistent framework, especially in data-lacking, heterogeneous regions (Han et al., 2021; Winckler et al., 2018). Many research papers have used numerical models to systematically interpret the energy cycle and hydrological cycle, contributing to a better understanding of the LUCC in water-energy processes (Chen and Dirmeyer, 2019; Llopart et al., 2018).

The Community Land Model (CLM), in which each grid cell is composed of multiple land use/cover, represents well under different land use/cover and LUCC regions (Li, 2021; Meier et al., 2018; Xu et al., 2020; Lawrence et al., 2019), which would effectively simulate changes of water-energy processes response to LUCC.

To investigate the impacts of LUCC, many studies focused on land surface temperature (LST) and evapotranspiration (ET), which both are extremely sensitive to LUCC and provide important information related to extreme events and water resources

management (Chen and Dirmeyer, 2018; He et al., 2020; Li et al., 2015; Wang et al., 2020). Studies have suggested that the impacts of LUCC on LST vary mainly resulting from the competition among different biogeophysical characteristics, like surface albedo and surface roughness (Burakowski et al., 2018; Cherubini et al., 2018; Davin and Noblet-Ducoudré, 2010; Li et al., 2015), and LUCC alters the redistribution of moisture flux and energy balance through biogeophysical characteristics, which differ for LUCC types and spatial variability, and then leads to the impacts on ET (Das et al., 2018; Li et al., 2017; Ning

et al., 2017; Winckler et al., 2017). Additionally, the diurnal cycle has been widely adopted to clearly show the discrepancy of fluxes distribution including soil residual heat fluxes and latent heat fluxes, representing temperature and ET, in different land use/cover and explicitly explain how biogeophysical characteristics in the LUCC process affect energy and water cycle (Breil



et al., 2020; Kueppers and Snyder, 2011). However, the spatially averaged impacts of LUCC or the impacts of single types of LUCC in LST and ET have been studied extensively (Cherubini et al., 2018; Davin and Noblet-Ducoudré, 2010), few
researchers quantified and attributed the spatially averaged impacts to the synergy of different LUCC types in complicated realistic conditions. Therefore, in this context, LST and ET are selected as representatives to quantify the synergy and respective impacts of different types of LUCC, which will help to explain the mechanisms of optimal land use/cover pattern. The agricultural pastoral ecotone in Northwest China (APENC), mainly interlaced by grasslands, croplands and bare land, is part of one of the largest agropastoral ecotones in the world (Li et al., 2018; Xue et al., 2019; Yang et al., 2021a). The land
surface vegetation has been experiencing large-scale changes over the last decades as a result of implemented policies, such as the "Grain for Green Project" and "Three-North Shelterbelt" (Cao et al., 2015; Wei et al., 2018; Liu et al., 2019). Those programs contributed to increasing vegetation (Wang et al., 2019b; Wu et al., 2013; Xue et al., 2019; Zhang et al., 2018) and then vegetation restoration led to increased soil moisture consumption (Yang et al., 2021a), reduction of runoff (Liang et al., 2015; Zhang et al., 2016), increased ET (Wang et al., 2019a) and decreased LST (Wang et al., 2020). However, some studies
pointed out that excessive re-vegetation being introduced caused negative effects like soil drying (Jia et al., 2017b; Zhang et al., 2018), indicating that incorporating a mixture of land use/cover into decision-making suitable for the APENC is urgently required. Previous studies in APNEC mainly focused on assessing changes in warming effects and water conservation (W) due to LUCC, and giving optimization suggestions based on changes in water conservation or delimiting the optimization area based on high W (Wang et al., 2020; Yang et al., 2021a; Zeng and Li, 2019; Jia et al., 2017a), while a couple of attempts about
optimized configuration have been done by only using an ecological coefficient in scenario simulation (Yang et al., 2020). Such optimizations lack comprehensiveness consideration within the local hydrological and climatic thresholds (warming effects and water conservation) standing perspective of water-energy processes. Additionally, the latest national ecological development projects plan to expand grasslands to 60 % in China and continue to convert bare and agricultural lands to grasslands or forests to improve ecosystem services in the APNEC from 2021 to 2035
(http://www.gov.cn/zhengce/content/2017-02/04/content_5165309.htm, Notice of The State Council on Printing and distributing the Outline of the National Land Plan (2016-2035), 2017; http://gi.mnr.gov.cn/202006/t20200611_2525741.html, Major projects for ecological protection and restoration support systems, 2019). However, this plan that expands grasslands to 60 % has not been robustly tested, and little was done to propose the proper percentages of croplands and bare land suitable for the APENC under the government plan. Thus, two criteria, LST and W (Bai et al., 2019; Findell et al., 2017; Wang et al.,
2021b), are considered as hydroclimatic thresholds to pursue proper land management plans suitable for APNEC, for the first time, within scenarios simulation of different vegetation restoration under CLM 5.0.

This study intends to bridge the current research gap in processed regional land use/cover patterns and propose the proper percentages of croplands and bare land suitable for the APENC under the government plan. The objectives of this study are 1) to quantify synergy and respective impacts of different types of LUCC, and 2) to find the proper mixture of land use/cover in
the APENC for 2035.



## 2 Materials and Methods

### 2.1 Study area

The APENC is located in a semi-arid region of northwestern China and has gotten a lot of attention owing to being an idealized location to study water-energy processes' response to landscape modification. It lies between 36.816 to 40.194 °N and 106.228 to 110.903 °E (Fig. 1), covering 77,513 km², at an elevation of 915-1947 m above mean sea level with an annually averaged temperature of 7.0 to 9.0 °C, an annual averaged relative humidity of 13 %, and annual precipitation of 250 to 450 mm with most of it falling in the summer (Xu et al., 2020; Yang et al., 2021a). The dominant vegetation types in sequence are grasslands, bare land and croplands. The study area is a climatic and ecological transition belt historically developed by agricultural cultivation and animal husbandry, highly sensitive to changes in human activities and background climate (Tan et al., 2020; Wei et al., 2018; Xue et al., 2019).

**Figure 1 (a) DEM of agro-pastoral ecotone of northwestern China and locations of the in situ observation stations. (b) Land use/cover map of the study area in 2000. (c) Land use/cover map of the study area in 2015.**



**2.2 Datasets**

The surface land use/cover dataset covers the study area with a 30 m ×30 m resolution. The 2000 and 2015 were selected to represent the land use/cover before and after the vegetation restoration project. The land use/cover dataset over the APENC contains 8 land use/cover types, including shrublands, grasslands, croplands, urban, barren land, water bodies, evergreen needleleaf forests and deciduous broadleaf forests, which correspond to land use/cover types of CLM's input surface data. The data of rainfed croplands and irrigated croplands were calculated as the ratio of irrigated lands to cultivated lands from

yearbooks of Shanxi, Ningxia and Erdos (Xu, 2018; Yang, 2021), shown in Table 1. The China meteorological forcing dataset (CMFD, http://data.tpdc.ac.cn), with a 3-hour time step and a horizontal spatial resolution of 0.1 degrees, covers the period 1979 to 2018 (Yang and He, 2016). The Dataset of soil properties for land surface modeling over China (http://data.tpdc.ac.cn) with 30×30 arc-second resolution includes sand content, clay content, soil organic matter and bulk density (Shangguan and Dai, 2013).

**Table 1 Percent of rainfed croplands and irrigated croplands on the APENC (Xu et al. 2018; Yang et al. 2021a).**

|      | Rainfed croplands (%) | Irrigated croplands (%) |
|------|------|------|
| 2000 | 61.30 | 38.70 |
| 2015 | 46.48 | 53.52 |

The six in situ observation stations were devised and established in 2016. Two Yanchi sites for croplands and grasslands, site 18, site 20 and site 39 for grasslands, and site 42 for croplands. The sampling locations are shown in Fig. 1 and Table 2. Their latitude, longitude and elevation were determined by a GPS receiver in the field survey. Soil temperatures and moisture have been being recorded every half hour since August 2016 using ECH20 senor to record 0-5 cm, 5-10 cm, 10-15 cm, 15-30 cm

and 30-50 cm soil layers. The MODIS LST (https://lpdaac.usgs.gov/dataset_discovery/modis) with 0.05 ° spatial resolution was used to validate LST over the domain, including daytime and nighttime (Wan et al., 2015). ET and net radiation were validated over the domain by two sensing products of GLASS (http://glass-product.bnu.edu.cn/): ET with 8-day temporal resolution and 0.05 ° spatial resolution and surface all-wave daily net radiation with daily temporal resolution and 0.05 ° spatial resolution (Guo et al., 2020; Yao et al., 2014).

**Table 2 Information of in situ sites on the APENC.**

| Site | Latitude (°E) | longitude (°N) | Altitude(m) | Underlying surface type |
|------|------|------|------|------|
| Yanchi grass | 37.9691 | 107.3851 | 1333 | Grasslands |
| Yanchi crop | 37.9690 | 107.3853 | 1333 | Croplands |
| 18 | 38.90 | 107.44 | 1222 | Grasslands |
| 20 | 39.17 | 108.38 | 1456 | Grasslands |
| 39 | 38.74 | 109.53 | 1264 | Grasslands |
| 42 | 38.19 | 108.97 | 1206 | Croplands |

A list of input datasets and validation datasets with their product name and support resources is presented in Table 3. For the convenience of model validation, we interpolated all data into 0.1 ° grids coincident with the spatial resolution of the model output.

**Table 3 List of inputs dataset and validation dataset.**



| | Variable | Product and support resource |
|---|---|---|
| Input data | land use/cover | Resources and Environment Data Center |
| | downward shortwave | |
| | longwave radiation | |
| | near-surface wind speed | |
| | near-surface temperature | CMFD by National Tibetan Plateau Data Center |
| | near-surface air specific humidity | |
| | near-surface air pressure | |
| | precipitation rate | |
| | sand content | |
| | clay content | Dataset of soil properties for land surface modeling by |
| | soil organic | National Tibetan Plateau Data Center |
| | bulk density | |
| Validation data | soil temperature | In situ observation |
| | ET | |
| | LST | MODIS by NASA |
| | ET | GLASS by National Earth System Science Data Center |
| | net radiation | |


## 2.3 Model description and experimental design

CLM5.0, developed by the National Center for Atmospheric Research (NCAR) and serving as the land surface component of the Community Earth System Model (CESM, http://www.cesm.ucar.edu/models/cesm2/), is a land surface model including biogeophysical and biogeochemical processes (Lawrence et al., 2019). In CLM5.0, each grid cell can have different numbers

of land units, including vegetated, crop, lake, urban and glacier. The vegetated land unit is divided into 16 plant functional types (PFTs) in the SP compset (Bonan et al., 2002; Lawrence et al., 2019). Details of the latest CLM, which was newly adopted in this study, can be found in the technical description of version 5.0 (http://www.cesm.ucar.edu/models/cesm2/land/CLM50_Tech_Note.pdf).

Because we need to represent the local crop of APNEC in CLM5.0, we modified the parameters of the C3 Unmanaged Crop

in the SP compset and then regarded it as corn. According to the local corn in the APNEC, the modifications in this study includes: LAI is 0 as a managed crop in the non-growing season, the canopy height of corn is 1.65 m by field gauge from 2017 to 2018, the C4 photosynthetic pathway because corn is a C4 plant, and stem area index (SAI) equal 0.1*leaf area index (LAI). The entire domain was produced in CLM5.0 with 40 x 50 horizontal grid cells with a spacing of 0.1 ° and each grid was composed of percentages of multiple land use/cover. Spin-up time reaching equilibrium was strictly constrained by $|Var_{n+1} -$

$Var_n| < 0.001*|Var_n|$ (Cai et al., 2014; Yang et al., 1995), where Var is each of the variables for the spin-up and n is the year for spin-up time. Soil moisture needed the longest memories according to Han et al. (2021), so soil moisture was selected as the constrained variable (Fig. 2). We cycled the atmosphere forcing of 1979-2018 twice to run spin-up. Thus, the results in 2000 and 2015 reached equilibrium and were used for analysis.





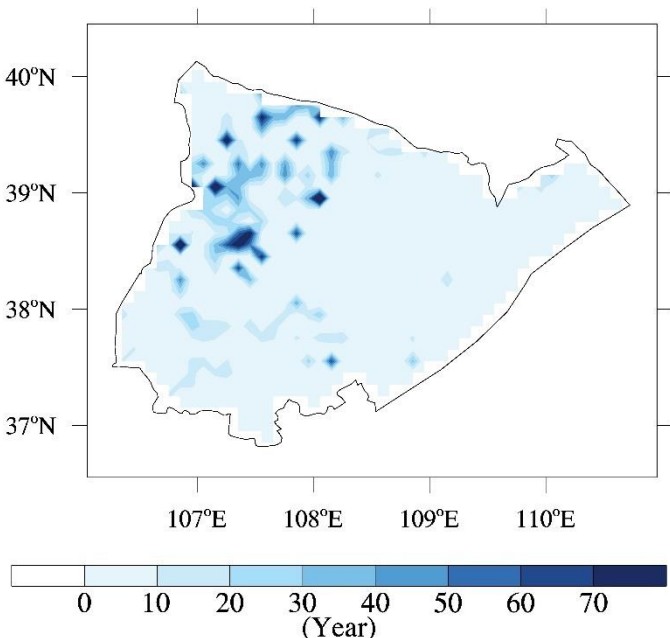

**Figure 2 Spin-up time for the study area.**

A suite of numerical simulations is described in Table 4 to evaluate the CLM5.0 and explore the impacts of LUCC. Firstly, single-point simulations with extreme single land cover/use were used to compare with in situ observations to evaluate the performance of CLM5.0 under different land use/cover. The CN2000 and CN2015 simulated the actual land surface and atmosphere forcing and then were used to assess the accuracy of CLM5.0 in the entire domain. Then, the impacts of LUCC were examined by the differences between EXP2000 and CN2015, isolating impacts caused by LUCC from 2000 to 2015(Wang et al., 2020). In the EXP_bare and EXP_crop scenarios, the bare land and croplands were extended to 100 %, respectively. Subsequently, in the EXP_grass scenario, grasslands were set to 100 % to replace bare land and croplands (Cherubini et al., 2018). In this way, EXP_grass, EXP_bare and EXP_crop were respectively simulated with extreme land use/cover to further analyse the impacts of different types of LUCC. Additionally, two sensitive experiments were simulated to examine the role of vegetation biogeophysical characteristics. The leaf and stem area index (LAI + SAI) of grasslands were replaced by crop in Yanchi_laisai, while canopy height in Yanchi_height (Breil et al., 2020).

**Table 4 List of numerical simulations.**

| Experiment | Region/points | Land use/land cover | Atmospheric Forcing | Grid |
|---|---|---|---|---|
| Yanchi_grass | Yanchi | grasslands | 2015-2018 | 0.0001 ° |
| Yanchi_crop | Yanchi | croplands | 2015-2018 | 0.0001 ° |
| 18_grass | 18 | grasslands | 2015-2018 | 0.0001 ° |
| 20_grass | 20 | grasslands | 2015-2018 | 0.0001 ° |
| 39_grass | 39 | grasslands | 2015-2018 | 0.0001 ° |
| 42_crop | 42 | croplands | 2015-2018 | 0.0001 ° |



| CN2000 | Domain | 2000 | 2000 | 0.1 ° |
| CN2015 | Domain | 2015 | 2015 | 0.1 ° |
| EXP2000 | Domain | 2000 | 2015 | 0.1 ° |
| EXP2015 | Domain | 2015 | 2000 | 0.1 ° |
| EXP_grass | Domain | Grasslands | 2015 | 0.1 ° |
| EXP_bare | Domain | Bare land | 2015 | 0.1 ° |
| EXP_crop | Domain | Croplands | 2015 | 0.1 ° |
| Yanchi_laisai | Yanchi | Yanchi | 2015 | 0.0001 ° |
| Yanchi_height | Yanchi | Yanchi | 2015 | 0.0001 ° |



## 2.4 Model evaluation

Three metrics were adopted to assess the performance of CLM5.0 in this study (Wang et al., 2019a; Zhang et al., 2021). correlation coefficient (R), absolute error (BIAS) and root mean squared error (RMSE) calculated following Eq. (1), Eq. (2) and Eq. (3):

$$R = \frac{\sum_{i=1}^{n}(X_i^{obs} - \bar{X}_i^{obs})(X_i^{sim} - \bar{X}_i^{sim})}{\sqrt{\sum_{i=1}^{n}(X_i^{obs} - \bar{X}_i^{obs})^2}\sqrt{\sum_{i=1}^{N}(X_i^{sim} - \bar{X}_i^{sim})^2}} \tag{1}$$

$$BIAS = \frac{1}{n}\sum_{i=1}^{n}(X_i^{sim} - X_i^{obs}) \tag{2}$$

$$RMSE = \sqrt{\frac{1}{n}\sum_{i=1}^{n}(X_i^{obs} - X_i^{sim})^2} \tag{3}$$

where $X_i^{obs}$ and $X_i^{sim}$ are the observed and simulated values for daily in in situ soil temperature, monthly in in situ ET, daily in regional LST and net radiation, and 8 days in regional ET (*i =1, 2, 3, …, n*), respectively, depending on the temporal scale of validation datasets. $\bar{X}_i^{obs}$ and $\bar{X}_i^{sim}$ are the averaged observed and simulated values during the simulated period, respectively. Previous work had validated soil moisture of output of CLM5.0 under grasslands and croplands in the APENC compared to in situ observation (Li, 2021). Here, we evaluated the simulated soil temperature in grasslands and croplands, which was in good agreement with in situ observation (Fig. 3). The R for Yanchi grass, Yanchi crop, 18, 20, 39 and 42 were 0.98, 0.98, 0.99, 0.96, 0.97 and 0.96, respectively. The BIAS for Yanchi grass, Yanchi crop, 18, 20, 39 and 42 were -1.09, -1.24, -0.85, -0.84, 0.44 and 0.09 ℃, respectively. The RMSE for Yanchi grass, Yanchi crop, 18, 20, 39 and 42 were 2.68, 2.07, 2.12, 3.28, 2.45 and 2.73 ℃, respectively. All single-point simulations at five depths showed high R (>0.95), low BIAS (< ± 1.71 ℃) and RMSE (<3.88 ℃). As shown in Fig. 4, the R, BIAS and RMSE between simulated and observed ET in the Yanchi station were 0.93, 15.52 mm month[-1] and 17.10 mm month[-1], respectively. Fig. 5 shows the spatiotemporal R between simulated LST, net radiation and ET and multiple validation datasets (MODIS, GLASS) in the entire domain. It can be seen that the R for LST, net radiation and ET were 0.96, 0.84 and 0.83, respectively. Although there was little bias in the water-energy processes of CLM5.0 due to parameterization (Deng et al., 2020; Luo et al., 2020; Ma et al., 2021), LUCC effects suppressed model uncertainty due to parameterization (Tölle et al., 2018). Therefore, the CLM5.0 can present complicated realistic LUCC in the APENC.





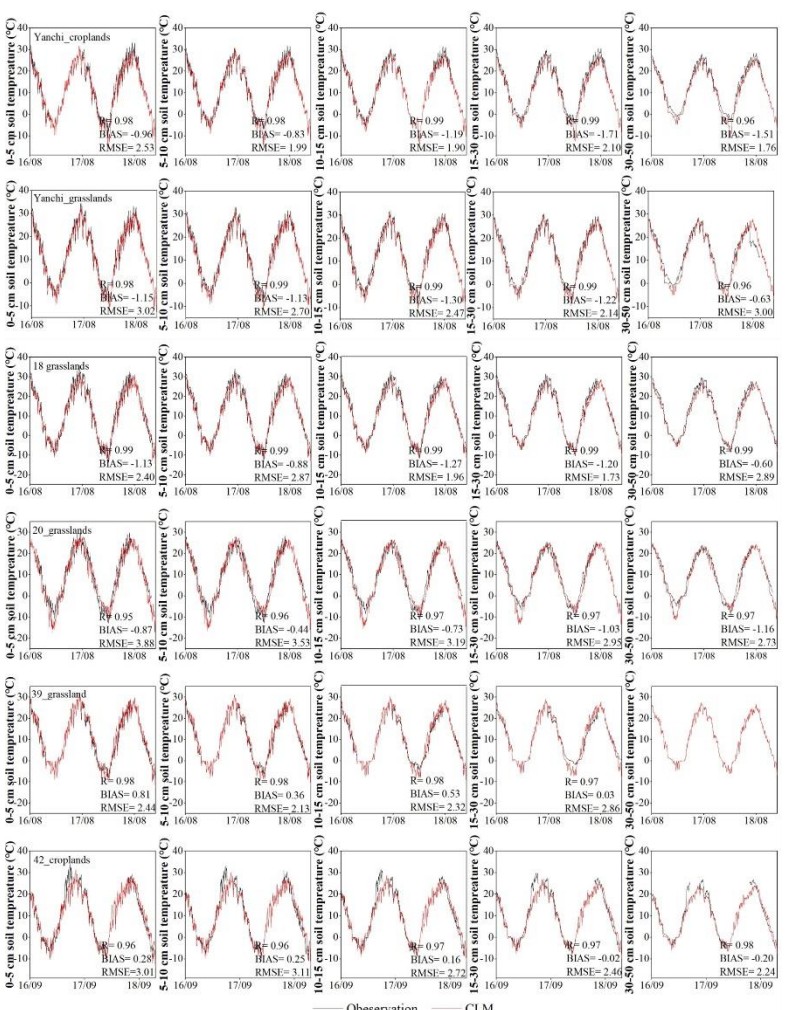

200

**Figure 3 Model simulations (red line) versus observations (black line) of the daily soil temperature at different depths (0-5, 5-10, 10-15, 15-30 and 30-50 cm) under different land use/cover types in Yanchi, 18, 20, 39 and 42 stations.**

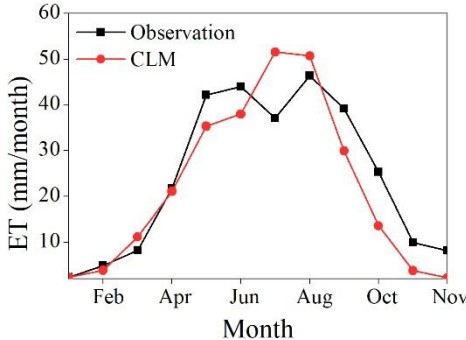

**Figure 4 Model simulations (red line) versus observations (black line) of monthly ET in Yanchi.**



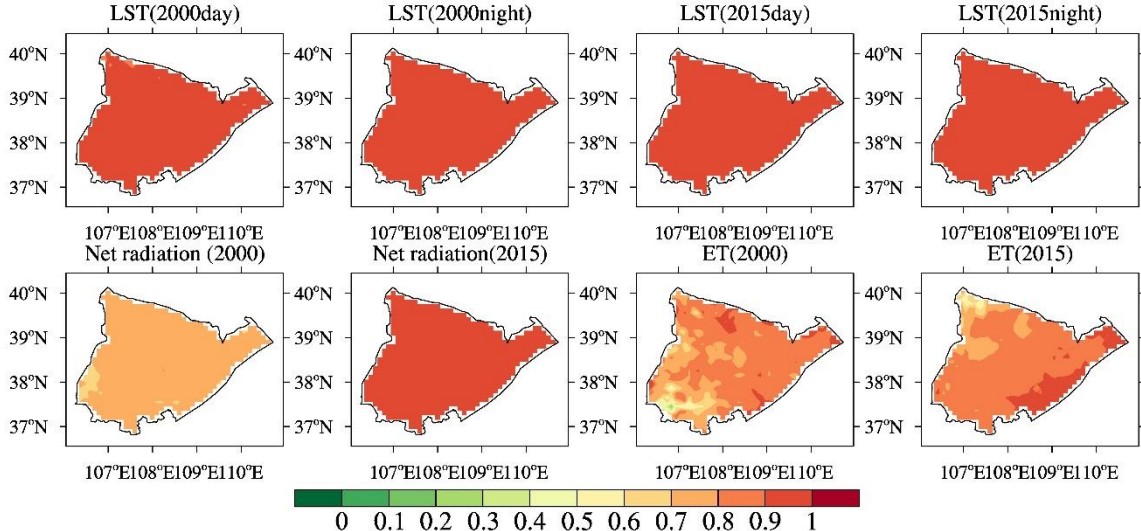

**Figure 5 The correlation coefficients of simulations and validation datasets (MODIS, GLASS) in 2000 and 2015.**

## 2.5 Criteria of appropriate land use/cover pattern

Considering the importance of warming impacts and the water conservation function, the proper mixture of land use/cover for 2035 depends on LST and W, which have been introduced as criteria for optimizing the ecosystem services from the perspective of energy and hydrology cycle (Bai et al., 2019; Zeng and Li, 2019; Wang et al., 2021a). The W was obtained from the water balance Eq. (4):

$$W = P - ET - Runoff \tag{4}$$

where $W$ is the annual water conservation (mm yr$^{-1}$). While $P$, $ET$ and $Runoff$ are annual precipitation (mm yr$^{-1}$), evapotranspiration (mm yr$^{-1}$) and runoff (mm yr$^{-1}$), respectively. P is the forcing data of CLM5.0 and other data are the results of CLM5.0, whose performance has been validated by Li (2021) and the last section.

## 3 Results

### 3.1 Impacts of LUCC in the APENC

#### 3.1.1 LUCC from 2000 to 2015

To quantify synergy and respective impacts of different types of LUCC, we need first to examine the local LUCC. From 2000 to 2015, grasslands, evergreen needleleaf forest and deciduous broadleaf forest and shrublands increased by 7.30 %, 0.17 %, 0.15 % and 0.07 %, respectively. Bare land and croplands decreased by 8.70 % and 0.20 %, severely. Overall, vegetation coverage in the APENC increased. There were four types of main LUCC: bare land to grasslands (BL to GRS) by 11.62 %, croplands to grasslands (CL to GRS) by 1.18 %, grasslands to bare land (GRS to BL) by 3.83 % and grasslands to croplands



(GRS to CL) by 1.03 %. The BL to GRS and CL to GRS were driven by vegetation restoration projects in the APENC. The
BL to GRS areas were mainly distributed in the northwestern APENC and scattered elsewhere, while GRS to BL took place
in the western APENC. The CL to GRS were principally distributed in the mid-western APENC and GRS to CL in the mid-
southern APENC.

Here, we focused on the main LUCC including BL to GRS, GRS to BL, CL to GRS and GRS to CL. Meanwhile, grid cells
that experienced intense single LUCC type changes ≥ 15 % (Winckler et al., 2018) and other changes ≤ 15 % were selected as
representatives for further analysis (Fig. 6).

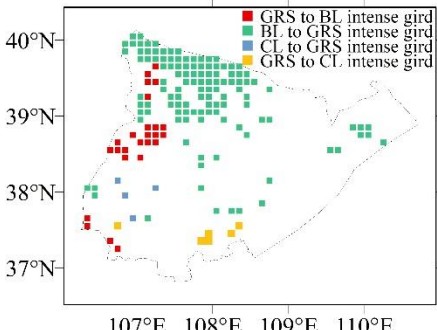

**Figure 6 Intense LUCC region with BL to GRS, GRS to BL, CL to GRS and GRS to CL.**

### 3.1.2 The impacts of LUCC over the domain

Here, we ran two experiments in CLM5.0 with two land use/cover of 2000 and 2015 under the same atmosphere forcing,
thereby eliminating the influence of external conditions and isolating the impacts of LUCC on LST. Fig. 7 shows the spatial
and seasonal distribution of the temperature differences between CN2015 and EXP2000. The LUCC from 2000 to 2015
generally caused a cooling effect in large areas of the APENC, where the spatially averaged cooling was -0.06 ± 0.15 ℃ (mean
± one standard deviation) as a result of increased vegetation coverage. Areas towards the eastern part of the APENC showed
a weak effect owing to the slight LUCC in the east (Fig. 6). Seasonally changes were -0.06 ± 0.15 ℃ in spring (MAM: March
& April & May), -0.12 ± 0.22 ℃ in summer (JJA: June & July & August), -0.06 ± 0.14 ℃ in autumn (SON: September &
October & February) and -0.02 ± 0.17 ℃ in winter (DJF: December & January & February).

Same as LST, we only considered the changes in ET directly caused by LUCC by isolating the effects of the LUCC. The
spatial and seasonal distribution of the ET differences between CN2015 and EXP2000 was shown in Fig. 8. The LUCC from
2000 to 2015 generally caused increased ET in large areas of the APENC, where the difference was 9.70 ± 19.04 mm yr$^{-1}$ as a
245 result of increased vegetation coverage. Seasonally changes were 1.93 ± 4.41 mm season$^{-1}$ in spring, 6.53 ± 11.67 mm season$^{-1}$ in summer, 1.16 ± 3.99 mm season$^{-1}$ in autumn and 0.07 ± 0.88 mm season$^{-1}$ in winter. The LUCC mainly affected ET in
summer, but this trend was weak in autumn and non-existent in winter.



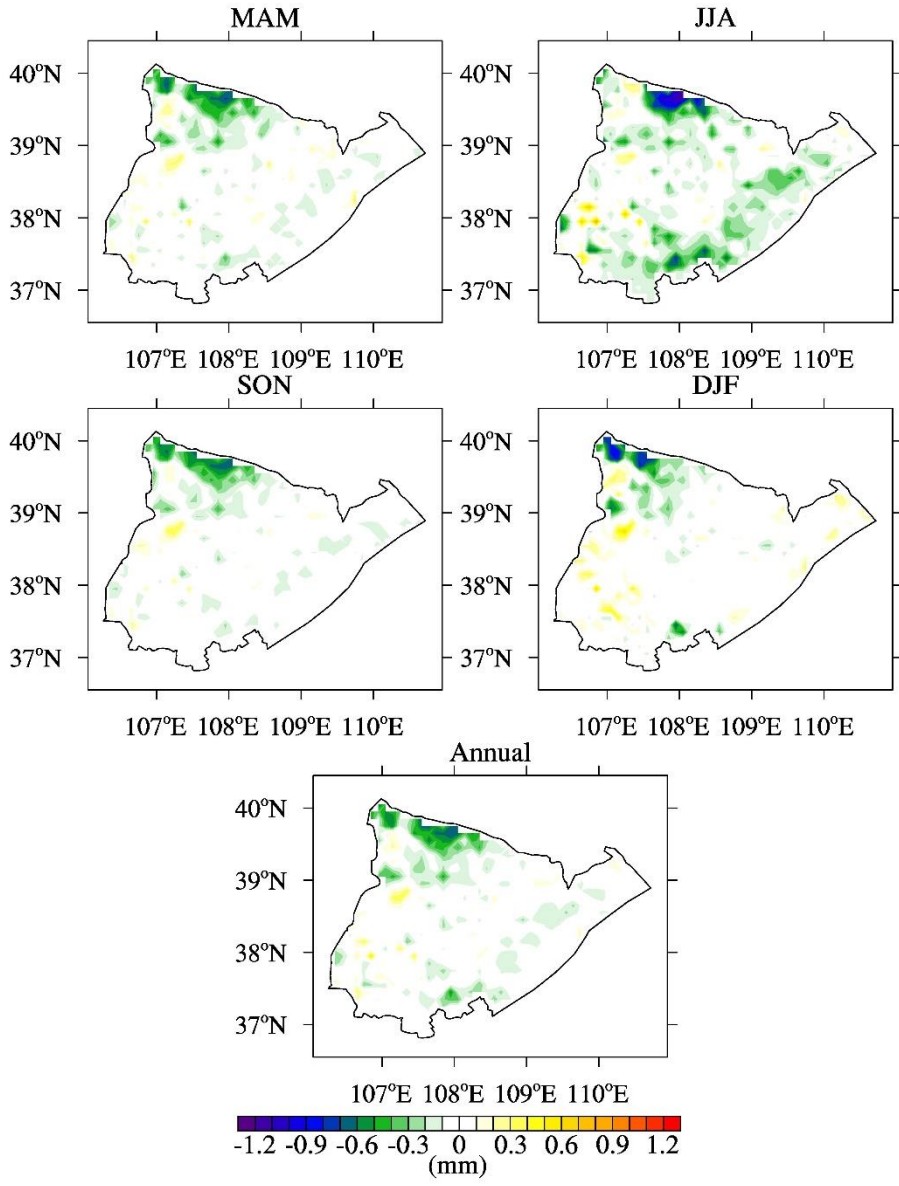

**Figure 7 Differences in spatially averaged LST between the simulations with 2000 and 2015 land use data (CN2015 - EXP2000) during MAM, JJA, SON, DJF and annual.**





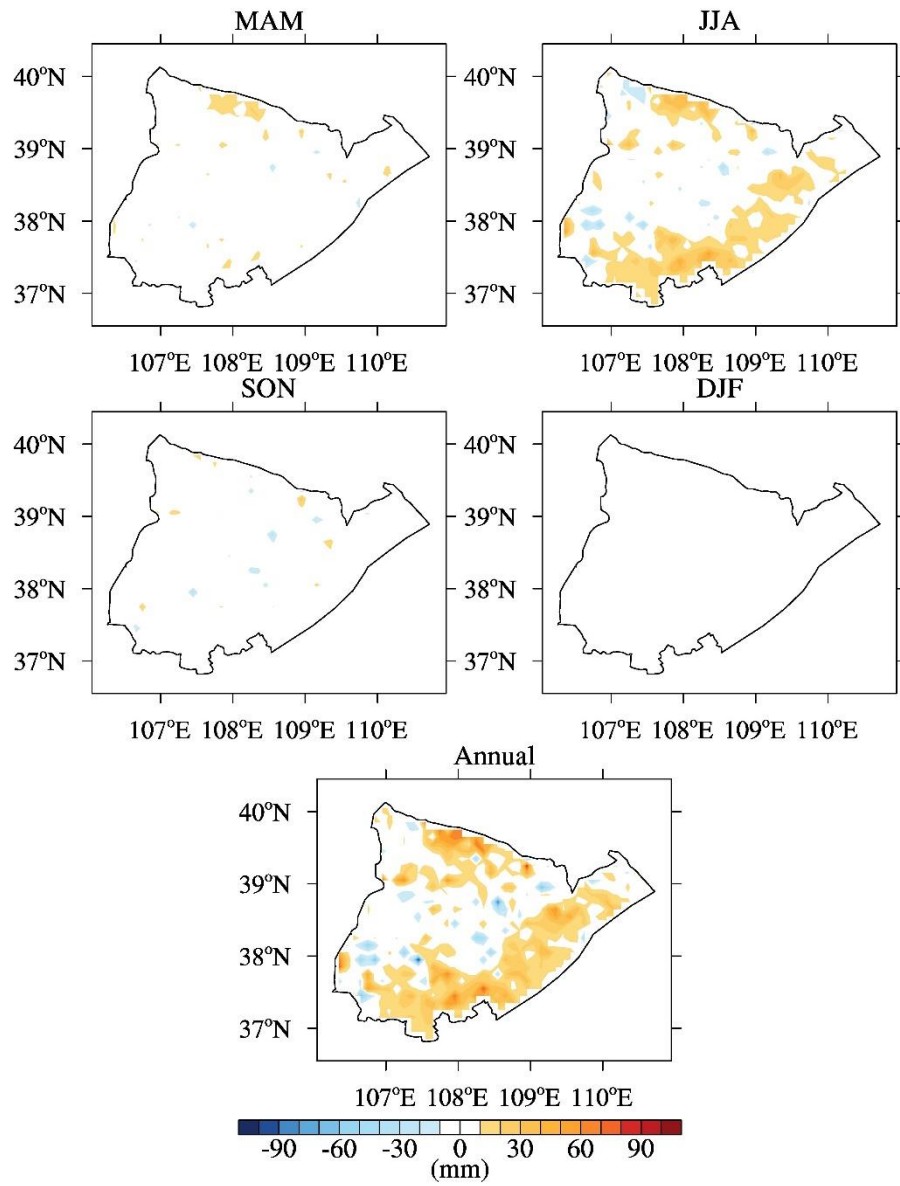

**Figure 8 Differences in spatially averaged ET between the simulations with 2000 and 2015 land use data (CN2015 - EXP2000) during MAM, JJA, SON, DJF and annual.**

### 3.1.3 Effects of different LUCC types

255   While different types of LUCC contributed to different effects and eventually led to the synergy effects over the domain. To understand the different effects of different types of LUCC, BL to GRS and CL to GRS as two main types of vegetation restoration projects could be carried out through two idealized scenarios which means one continent with maximized bare land



is turned into grasslands, and the other continent with maximized croplands is turned into grasslands (Arora and Montenegro, 2011; Cherubini et al., 2018). A detailed description of the scenarios can be found in Table 4. Analyses of water-energy response to BL to GRS were conducted in BL to GRS and GRS to BL intense grid cells (143 grids, Fig. 6), where bare land and grasslands realistically exist and constantly change back and forth. Similarly, analyses of CL to GRS were conducted in CL to GRS and GRS to CL intense grid cells (10 grids, Fig. 6), where crops and grass could grow and convert with each other. The Fig. 9 shows opposing impacts response to two types of vegetation restoration. The BL to GRS reduced the LST with -0.17 ℃ in annually averaged difference. On the contrary, CL to GRS led to an increase in LST with an annually averaged difference of 0.96 ℃. For BL to GRS scenarios, averaged seasonal cooling differences were -0.15 ℃, -0.74 ℃ and -0.66 ℃ in spring, summer and autumn, respectively, but warmer in winter with 0.89 ℃. Temperature impacts for CL to GRS showed a warm effect with a more dramatic variation, with averaged seasonal differences of 0.08 ℃, 2.52 ℃, -0.07 ℃ and 1.30 ℃ in spring, summer, autumn and winter, respectively. Annual changes in ET were 53.32 and -184.42 mm yr$^{-1}$ for BL to GRS and CL to GRS, respectively. The differences in ET for BL to GRS were 15.67, 23.77, 11.99 and 2.37 mm season$^{-1}$ in spring, summer, fall and winter, respectively. Conversely, the differences in ET for CL to GRS were -34.95, -128.76, -23.48 and 2.76 mm season$^{-1}$ in spring, summer, fall and winter, respectively. For CL to GRS (Table 5) and BL to GRS (Table 6), the surface albedo was the most sensitive factor and presented a significant correlation with LST and ET in summer and winter. The correlation coefficients in Table 5 further indicated that LAI + SAI was the most sensitive factor in influencing LST and ET for CL to GRS throughout the year.

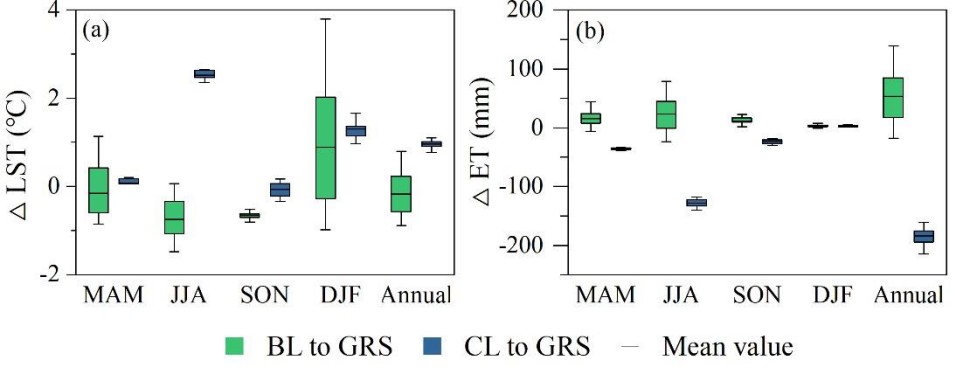

**Figure 9 Seasonal changes in LST and ET as box plots for BL to GRS (Exp_grass-Exp_bare) and CL to GRS (Exp_grass-Exp_crop).**

**Table 5 Relationships between differences in LST and ET and surface albedo, surface roughness, LAI + SAI, aerodynamic resistance, vegetation displacement height, leaf stomatal resistance and vapor pressure, respectively, in the intense LUCC region ( EXP_grss - EXP_crop).**

|  | Δ LST | | | | | Δ Latent heat flux/Δ ET | | | | |
|---|---|---|---|---|---|---|---|---|---|---|
|  | MAM | JJA | SON | DJF | year | MAM | JJA | SON | DJF | year |
| Δ surface albedo | 0.22 | 0.35 | -0.07 | -0.47 | -0.18 | 0.01 | -0.36 | -0.10 | -0.38 | -0.12 |
| Δ surface roughness | 0.13 | -0.19 | -0.06 | -0.07 | 0.01 | -0.27 | 0.07 | -0.27 | 0.21 | -0.27 |
| Δ LAI + SAI | -0.20 | -0.19 | -0.15 | 0.02 | -0.21 | 0.33 | 0.07 | 0.30 | 0.24 | 0.38 |





| | | | | | | | | | |
|---|---|---|---|---|---|---|---|---|---|
| Δ aerodynamic resistance | 0.01 | -0.04 | -0.05 | -0.33 | -0.12 | -0.03 | -0.05 | -0.05 | -0.19 | -0.04 |
| Δ vegetation height | 0.09 | -0.22 | -0.10 | -0.03 | -0.04 | 0.24 | 0.07 | -0.21 | 0.06 | -0.20 |
| Δ leaf stomatal resistance | 0.11 | 0.19 | 0.08 | -0.03 | 0.10 | -0.15 | -0.11 | 0.17 | 0.07 | -0.03 |

280  **Table 6 Relationships between differences in LST and ET and surface albedo, surface roughness and aerodynamic resistance, respectively, in the intense LUCC region ( EXP_grass - EXP_bare).**

| | Δ LST | | | | | Δ Latent heat flux/Δ ET | | | | |
|---|---|---|---|---|---|---|---|---|---|---|
| | MAM | JJA | SON | DJF | year | MAM | JJA | SON | DJF | year |
| Δ surface albedo | 0.01 | 0.33 | -0.02 | -0.30 | -0.10 | -0.14 | -0.51 | -0.08 | -0.34 | -0.18 |
| Δ surface roughness | 0.06 | -0.22 | -0.04 | 0.15 | -0.05 | -0.21 | 0.13 | -0.27 | 0.38 | -0.24 |
| Δ aerodynamic resistance | -0.06 | -0.08 | 0.01 | -0.22 | -0.08 | -0.05 | 0.07 | -0.01 | -0.22 | -0.01 |

Further analysis was undertaken focusing on the opposing mechanism response between BL to GRS and CL to GRS. Complete diurnal cycles were only shown for summer and winter here, which were considered the most representative of all seasons.

**a. BL to GRS**

285  In the summer days of CLM5.0 simulations, LST showed cooling for BL to GRS (-0.74 ± 0.99 °C, Fig. 10h). For bare land, surface temperature equals ground temperature. For vegetation cover, the surface temperature is a calculation related to ground temperature and vegetation temperature (Lawrence et al., 2019). The ground temperature is decided by the amount of energy that is used to warm ground and soil, residual heat energy, resulting from the competition of the net radiative energy input and the sum of the turbulent heat fluxes (sensible + latent heat fluxes) (Breil et al., 2020). In Fig. 10f, differences in the ground

290  temperature for BL to GRS were relatively small (-0.05 ± 0.48 °C), so the reduced surface temperature for BL to GRS mainly was caused by a lower vegetation temperature of grass (Fig. 10g).

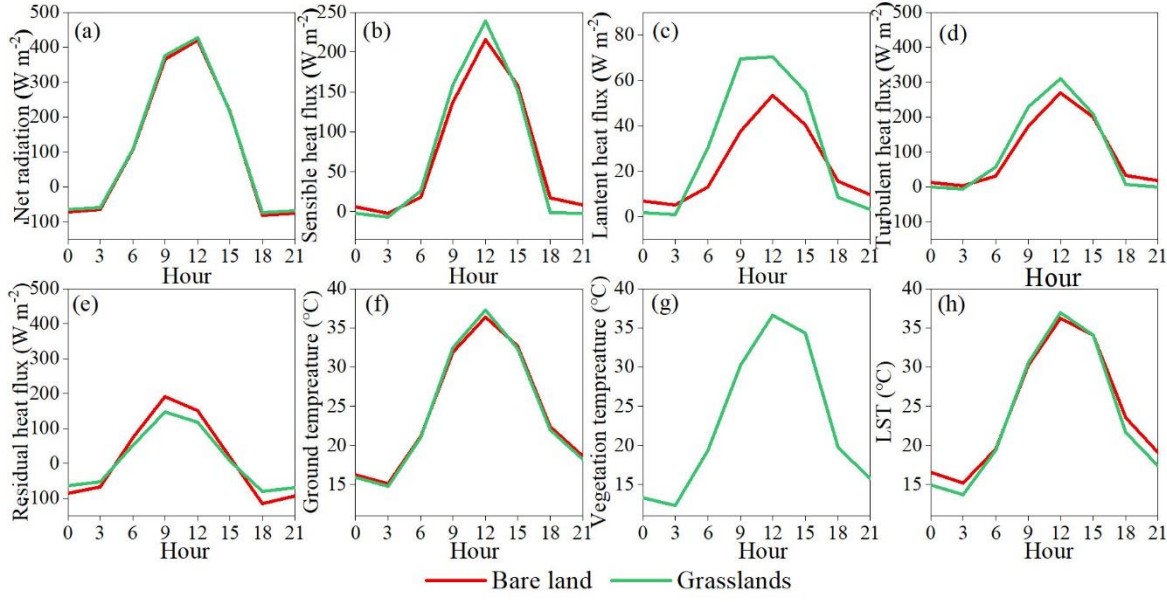



**Figure 10 Mean seasonal diurnal cycle in summer (Exp_bare & Exp_grass): (a) net radiation, (b) sensible heat fluxes, (c) latent heat fluxes, (d) turbulent heat fluxes, (e) residual heat fluxes (soil heat fluxes), (f) ground temperature, (g) vegetation temperature, (h) LST.**

In winter, LST increased by 0.89 ± 1.27 °C for BL to GRS. The lower grass surface albedo than bare land drove a significant reduction in surface albedo, leading to increased net radiation for BL to GRS. The increased sensible heat fluxes and latent heat fluxes were minimal (Fig. 11b, 11c), meaning that the increased turbulent term (about up to 32 W m$^{-2}$, Fig. 11d) was compensated by increased net radiation (about up to 52 W m$^{-2}$, Fig. 11a), suggesting the net shortwave radiation acted as the main term. Thus, the LST increased since residual heat increased (about up to 21 W m$^{-2}$, Fig. 11e).

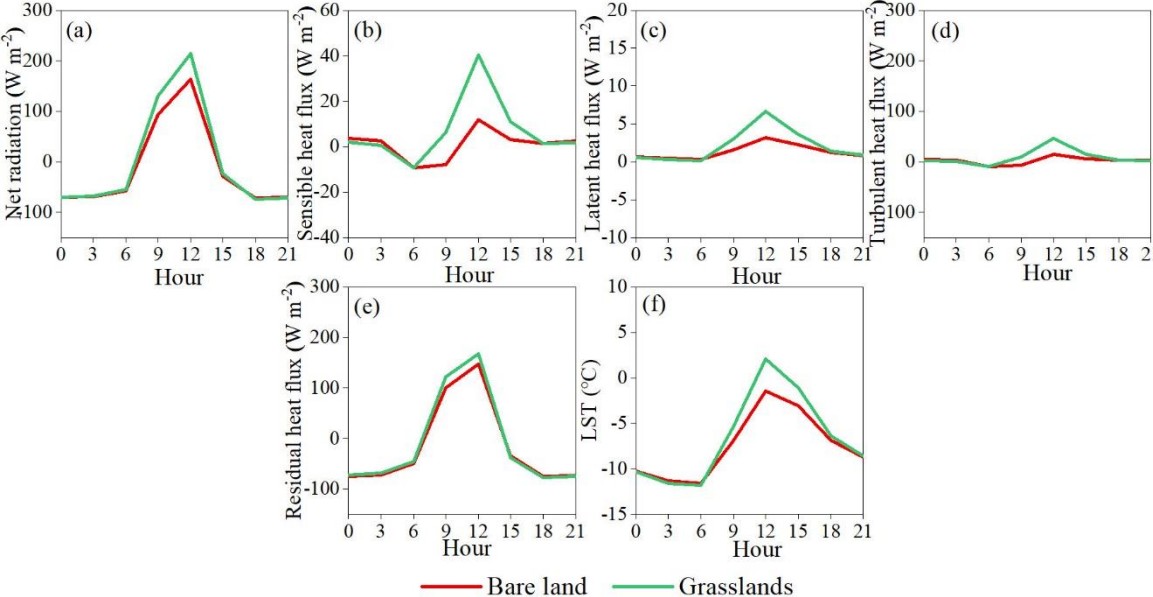

**Figure 11 Mean seasonal diurnal cycle in winter (Exp_bare & Exp_grass), (a)-(e) description same as Figure 10, but (f) for LST.**

**b. CL to GRS**

LST showed warming for CL to GRS (2.52 ± 2.35 °C, Fig. 12f). The net radiation decreased for CL to GRS (about -40 W m$^{-2}$ at daily maximum, Fig. 12a), due to the higher albedo values of grass compared to crop. The decreased turbulent energy fluxes (about -60 W m$^{-2}$ at daily maximum, Fig. 12d) into the atmosphere were decided by decreased latent heat fluxes (about -133 W m$^{-2}$ at daily maximum, Fig. 12c) rather than increased sensible heat fluxes (about 73 W m$^{-2}$ at daily maximum, Fig. 12b). Ultimately, the decreased net radiative energy input was compensated by a decreased sum of turbulent heat fluxes during the day. Thus, the result shows that the LST increased during the day as increased residual heat fluxes (about 32 W m$^{-2}$ at daily maximum, Fig. 12e). At night, the reversed residual ground heat energy hardly reduced the nocturnal LST. This was interpreted as the amount of energy increased at night was not enough to compensate for the larger temperature during the day (Breil et al., 2020).



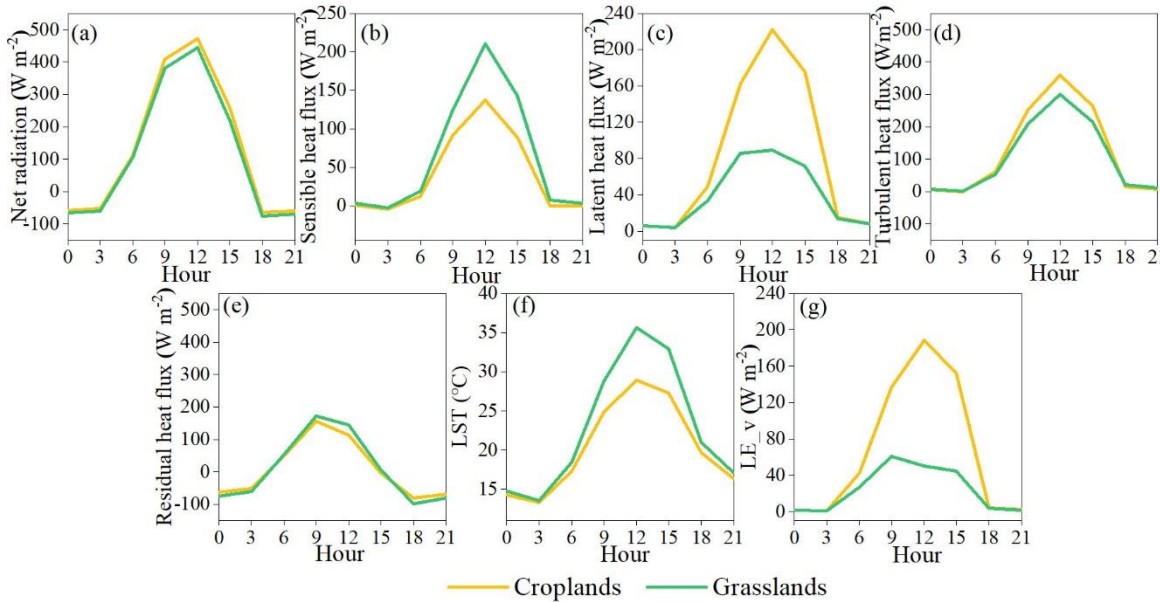

**Figure 12 Mean seasonal diurnal cycle in summer (Exp_crop & Exp_grass), description same as Figure 11, but (g) for vegetation latent heat fluxes (LE_v).**

In winter, the LST increased by 1.30 ± 0.38 °C for CL to GRS. There were no major differences between BL to GRS and CL to GRS owing to croplands having no vegetation in winter after being managed and being analogous to bare land after harvest in autumn in CLM.

## 3.2 Land use/cover pattern in the APENC

### 3.2.1 Spatiotemporal mixture of land use/cover

Aimed to explore the proper mixture of land use/cover, the mixtures of land use/ cover in 2000 and 2015 were analysed. There were different percentages representing the mixture of land use/cover in each grid. Here, we classified the mixture of land use/cover. To simplify the classification, only the grids that sum areas of grasslands, bare land and croplands greater than 90 % were selected, and then the ratio of three main types in each grid represented the mixture of land use/cover.

Fig. 13 shows the spatial-temporal heterogeneity of the mixture of land use/cover about three main types: grasslands, croplands and bare land, and each grid has a mixed land use /cover. The different impacts of vegetation restoration from 2000 to 2015 were represented in the terms of grids from CN2015 to EXP2000, shown in Table 7. Different effects of vegetation restoration resulted from different contributions of two main types of LUCC: grids of BL to GRS led to more cooling and drying, grids of CL to GRS led to more warming and moisture, which are in line with the section 3.1.3, however, a grid of BL to GRS and CL to GRS led to more warming and drying, which is due to the opposing offsetting impacts of CL to GRS and BL to GRS.



Therefore, unclear synergy effects of BL to GRS and CL to GRS as re-vegetation. A proper mixture of land use/cover in 2035 conversions from 2015 were explored in the next section.

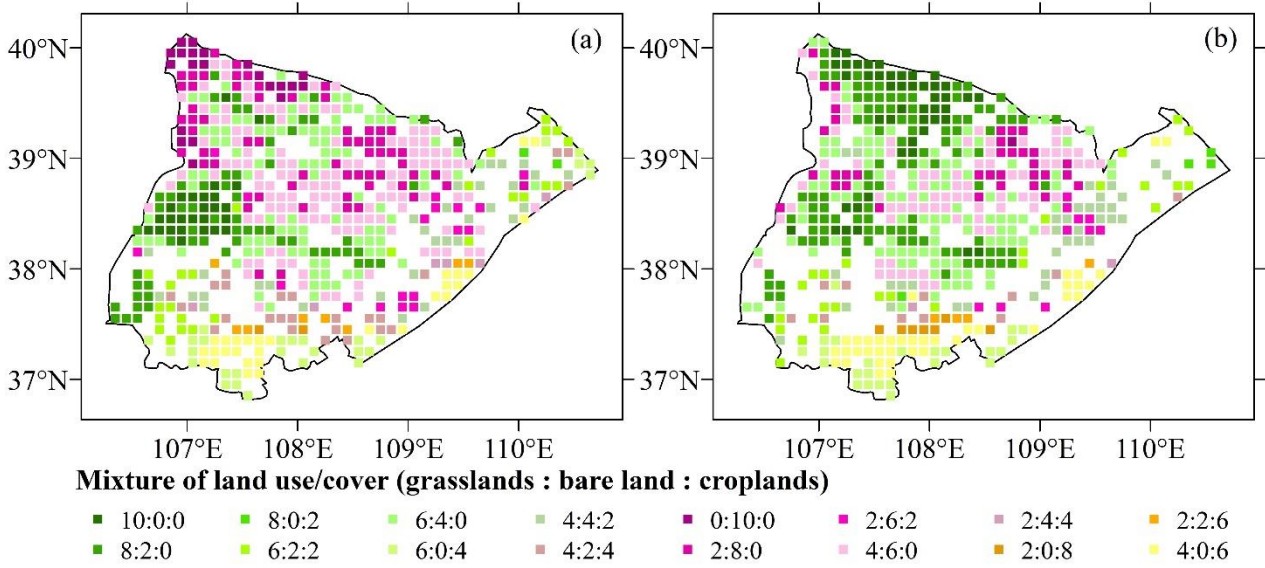

**Figure 13 The pattern of mixtures of land use/cover in 0.1 ° grids of the study area in 2000 (a), 2015 (b).**


**Table 7 The differences of LST and W as LUCC pattern change between 2000 and 2015 (CN2015 - EXP2000).**

| Change type | Mixture (grasslands: bare land: croplands) | grids | Δ LST(°C) | Δ W(mm yr⁻¹) |
|---|---|---|---|---|
| | 8:2:0 to 10:0:0 | 8 | -0.09 | -2.32 |
| | 6:4:0 to 10:0:0 | 9 | -0.26 | -20.51 |
| | 6:4:0 to 8:2:0 | 20 | -0.12 | -11.25 |
| | 4:4:2 to 6:2:2 | 5 | -0.08 | -16.39 |
| | 4:2:4 to 6:0:4 | 2 | -0.06 | -21.32 |
| | 0:10:0 to 10:0:0 | 9 | -0.63 | -44.78 |
| | 0:10:0 to 8:2:0 | 3 | -0.52 | -22.26 |
| | 0:10:0 to 6:4:0 | 3 | -0.46 | -11.41 |
| BL to GRS | 2:8:0 to 10:0:0 | 5 | -0.44 | 5.22 |
| | 2:8:0 to 8:2:0 | 6 | -0.51 | -24.41 |
| | 2:8:0 to 6:4:0 | 4 | -0.30 | -35.12 |
| | 2:6:2 to 4:4:2 | 6 | -0.14 | -29.68 |
| | 4:6:0 to 10:0:0 | 6 | -0.32 | -19.98 |
| | 4:6:0 to 8:2:0 | 12 | -0.30 | -13.42 |
| | 4:6:0 to 6:4:0 | 26 | -0.10 | -7.89 |
| | 2:4:4 to 4:2:4 | 1 | -0.12 | -26.65 |
| | 6:2:2 to 8:2:0 | 1 | 0.07 | 23.73 |
| CL to GRS | 6:0:4 to 8:0:2 | 2 | 0.07 | 1.77 |
| | 2:2:6 to 6:2:2 | 1 | 0.33 | 53.92 |
| BL to GRS and CL to GRS | 2:4:4 to 6:2:2 | 1 | 0.13 | -2.58 |





### 3.2.2 Proper mixture of land use/cover for future revegetation operations

#### a. Land use/cover scenarios based on the national ecological plan

The aim of the government plan for 2035 is 1) the grasslands of 60 %, and 2) re-vegetations by BL to GRS and CL to GRS

(http://www.gov.cn/zhengce/content/2017-02/04/content_5165309.htm, Notice of The State Council on Printing and distributing the Outline of the National Land Plan (2016-2035), 2017; http://gi.mnr.gov.cn/202006/t20200611_2525741.html, Major projects for ecological protection and restoration support systems, 2019). Meanwhile, the percent of grasslands, croplands and bare land in 2015 were respectively ~53 %, ~13 % and ~30 %. Thus, the grasslands will increase to 60 %, bare land will decrease from 30 %, and croplands will decrease from 13 % during 2015 to 2035. Different mixtures of land use/cover

were simulated to pursue a proper mixture of land use/cover in 2035. First, we set that the percentage of grasslands was 60 % in 2035. Then the percent of bare land and croplands, 13 % and 30 % respectively, was decreased in 2035 to meet the increase of grasslands and was set as the maximum in future scenarios. We ran all possibilities in that range, so five future scenarios, which consider computing time, to represent all possible vegetation restoration: the percent of grasslands, bare land and croplands were respectively 60 %, 21 % and 13 % in EXP_602113; 60 %, 23 % and 11 % in EXP_602311; 60 %, 25 % and

9 % in EXP_602509; 60 %, 27 % and 7 % in EXP_602707; 60 %, 30 % and 4 % in EXP_603004. The details are shown in Table 8.

**Table 8 numerical simulations for future scenarios.**

| Experiment | Land use/land cover | Forcing | Grid |
|---|---|---|---|
| EXP_602113 | grasslands60%, bare land 21%, croplands13% | 2015 | 0.1 ° |
| EXP_602311 | grasslands60%, bare land 23%, croplands11% | 2015 | 0.1 ° |
| EXP_602509 | grasslands60%, bare land 25%, croplands9% | 2015 | 0.1 ° |
| EXP_602707 | grasslands60%, bare land 27%, croplands7% | 2015 | 0.1 ° |
| EXP_603004 | grasslands60%, bare land 30%, croplands4% | 2015 | 0.1 ° |


#### b. Optimal land use/cover pattern

Here, we compared the difference between land use/cover future scenarios of 2035 and 2015 under the same atmosphere forcing in CLM5.0, thereby eliminating the influence of external conditions and isolating the impacts of LUCC in 2035. As is

shown in Table 9, both EXP_602113 and EXP_602311 led to the cooling surface while EXP_602509, EXP_602707 and EXP_603004 caused a warming surface in 2035. Additionally, EXP_602113 induced a drying while EXP_602311, EXP_602509, EXP_602707 and EXP_603004 led to high water conservation.

For sustainable ecological construction, there was a need that pursues an alternative proper mixture of land use/cover without augmenting warming and endangering future water availability, which means the proper mixture of land use/cover has lower

LST and bigger W than in 2015 (Arora and Montenegro, 2011; Bai et al., 2019; Wang et al., 2021b; Findell et al., 2017). Therefore, vegetation restoration strategies in the APENC should use the proper mixture of land use/cover as EXP_602311, in which grasslands, bare land and croplands approximately are 60 %, 23 % and 11 %, respectively. It indicates that approximately 6.9 % of bare land and 1.5 % of croplands, respectively, transformed into grasslands from 2015 to 2035. The LUCC from 2015 to EXP_602311 generally will cause more cooling and a slightly increased water conservation, as a result of proper vegetation



restoration. Otherwise, other scenarios will lead to more warming or more drying results in 2035, exacerbating drought in the APENC.

**Table 9 The spatially weighted averaged differences of LST and W as different vegetation restoration efforts from 2015 to 2035.**

|  | Δ LST (°C) | Δ W (mm yr$^{-1}$) |
|---|---|---|
| EXP_602113 | -0.04 | -4.39 |
| EXP_602311 | -0.01 | 0.86 |
| EXP_602509 | 0.02 | 6.09 |
| EXP_602707 | 0.05 | 11.34 |
| EXP_603004 | 0.09 | 19.25 |

## 4 Discussion

### 4.1 Role of vegetation biogeophysical characteristics

In CLM5.0, a dual-source land surface model, canopy stored energy is zero and is regarded as massless. Vegetation vapor pressure, temperature and latent heat fluxes are iteration calculations by the Newton-Raphson method, with high complexity related to several land surface parameters like surface albedo, roughness, LAI + SAI, aerodynamic resistance, vegetation height and leaf stomatal resistance (Lawrence et al., 2019). To better understand the complex processes, the impacts of CL to GRS were investigated further. Here, additional sensitivity simulations were performed with CLM5.0, in which canopy height

(Yanchi _height) and LAI + SAI (Yanchi_laisai) of grass were replaced respectively by crop while all other land surface parameters of grass remained unchanged. To save computed times, the sensitivity experiments were only conducted in the Yanchi station as the most representative for CL to GRS. As shown in Fig. 14b, the latent heat fluxes and LST of Yanchi_height and Yanchi_laisai were only slightly different from Yanchi_grass and hardly closed to Yanchi_crop. Moreover, the LAI + SAI and canopy height affected surface roughness and aerodynamic resistance (Fig. 14g, 14h). It means that complex processes

may not be simply adjusted with a single factor and other characteristics play indispensable roles. Future work in studying water-energy processes should combine with the interpretation distribution of the fluxes cycle shown in section 3.1.3, instead of simply considering the correlation between the variables and biogeophysical characteristics.





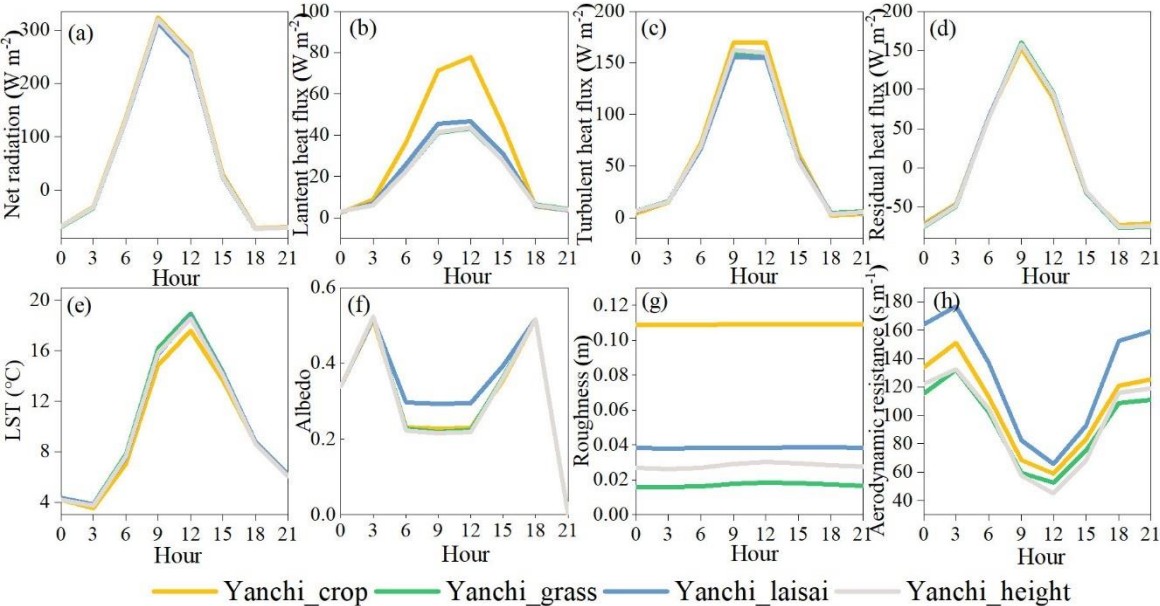

**Figure 14 Diurnal cycle (Yanchi_crop & Yanchi_grass & Yanchi_laisai & Yanchi_height, (a) net radiation, (b) latent heat fluxes, (c) turbulent heat fluxes, (d) residual heat fluxes (soil heat fluxes), (e) LST, (f) surface albedo, (g) surface roughness, (h) aerodynamic resistance.**

### 4.2 Soil properties in the input data of the model

Land surface processes are mainly presented as the interactions of soil-vegetation-atmosphere (Breil and Schädler, 2017). Soil properties serving as the lower boundary condition, such as thermal conductivity, porosity and hydraulic conductivity, are key parameters affecting soil moisture, soil temperature and soil heat fluxes which refer to the partitioning of water and energy (Yang et al., 2021b). Although the dataset of soil properties for land surface modeling over China provides higher precision soil properties than default values of CLM5.0 (Fig. 15), it is still worthwhile to explore the uncertainties of the modeling soil input dataset. Firstly, in Fig. 15, the dataset shows that sand context is less than 60 % and clay context is larger than 10 % in the northwest. However, as experiment data is shown in Table 10, northwest of the APENC is a desertified area where soil contains more mean sand and less mean clay (Duan et al., 2021; Liu et al., 2011; Xu, 2019), which implies that discrepancies exist between the soil dataset and realistic condition. Secondly, the conversion of land use/cover leads to soil properties change, especially soil organic matter, sand content and clay content (Celik, 2005; Su et al., 2021). The dynamic changes in soil properties under different land use/cover were not considered, and the same soil dataset was used before and after the LUCC. Most soil datasets contain soil properties as constant for a long time. Thus, the limitation was that soil properties could not change dynamically with LUCC, which might affect the simulated variables. Therefore, the accuracy of the soil dataset of the study area needs to be improved and a module that considers dynamic changes in soil parameters following LUCC is likely to be carried out in future research.



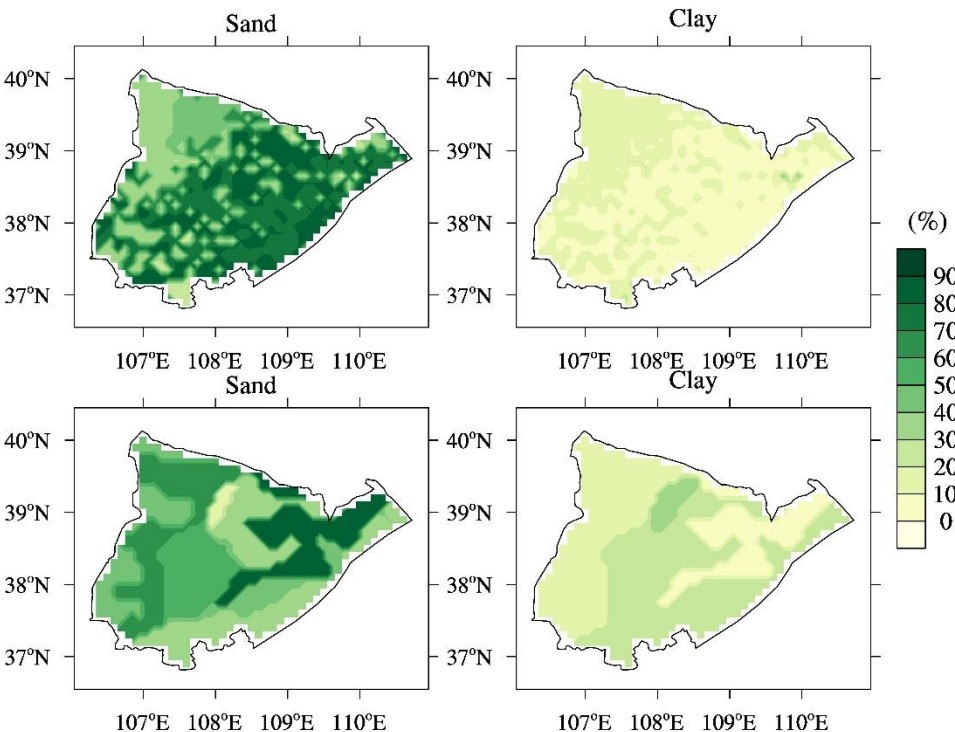

**Figure 15 Dataset of soil properties for land surface modeling over the APENC (upper panel) and default values (lower panel).**

**Table 10 Soil parameters for different soil layers in the experiment at MOE Key Laboratory of Western China's Environmental System at Lanzhou University.**

| Site | Sand/% | Clay/% |
|------|--------|--------|
| Yanchi_grass | 88.82 | 1.92 |
| Yanchi_crop | 90.66 | 1.13 |
| 18 | 87.01 | 2.30 |
| 20 | 90.59 | 1.02 |
| 39 | 96.70 | 0 |
| 42 | 96.89 | 0.10 |

## 4.3 Limitations of the study

Due to the limited in situ observations, this study validated the CLM5.0 with only 6 stations in the study region. Subsequently, the validated CLM5.0 was used to assess the proposed land use/cover scenarios. Future research needs to verify the proposed

scenarios with more diverse in situ observations before an appropriate land use pattern is selected to be implemented at regional scales.



## 5 Conclusions

This study first simulated and quantified the effects of LUCC on the associated surface water-energy impacts between 2000 and 2015 by CLM5.0, verified based on the in situ observations, in the APENC. Subsequently, five LUCC scenarios were
proposed, simulated and assessed to identify an optimal mixture of land use/cover based on the spatially averaged surface temperature and water conservation in the study region. The main results of this study are as follows:

1. From 2000 to 2015, the main LUCC was the reduction in croplands and bare land with an increase in grassland as a result of the implementation of ecological restoration projects.

2. The BL to GRS reduced LST while CL to GRS increased LST. The BL to GRS caused an increase in ET whereas CL to
GRS caused a decrease in ET. It led to a spatially averaged cooling surface and increased ET from 2000 to 2015 over the study area.

3. In-depth analysis of the LUCC pattern from 2000 to 2015 found that some grids showed warming or drying while a grid showed both drying and warming. This was due to the offsetting of opposing effects of BL to GRS and CL to GRS. Different mixtures of LUCC could lead to different results of re-vegetation projects, which indicates complicated synergy
effects of BL to GRS and CL to GRS as re-vegetation.

4. The Chinese government's long-term ecological plan targets the expansion of grasslands to 60 % by 2035 in the APENC. This study proposed and simulated the following five LUCC scenarios: the percent of grasslands, bare land and croplands were respectively 60 %, 21 % and 13 % in EXP_602113; 60 %, 23 % and 11 % in EXP_602311; 60 %, 25 % and 9 % in EXP_602509; 60 %, 27 % and 7 % in EXP_602707; and 60 %, 30 % and 4 % in EXP_603004. Assessing the five LUCC
scenarios by lower LST and higher W, the proper mixture of LUCC in the APENC in 2035 is approximately 60 %, grasslands, 23 %, bare land and 11 % croplands respectively, which will mitigate the drying and warming surface environment. The findings provide useful information to support land management policy/decision-making in the study region.

**Data availability**

The data that support the findings of this study are available upon reasonable request from the authors.



**Authorship contribution**

Yuzuo Zhu: Conceptualization, Methodology, Software, Validation, Formal analysis, Writing - original draft.

Xuefeng Xu: Data curation, Writing - Review & Editing.

**Declaration of competing interest**

We declare no competing interest.

**Acknowledgments**

The work is supported by the National Natural Science Foundation of China (Grants: 42030501, 41530752 and 91125010).
We thank Prof. Xin Jia and his team at Beijing Forestry University for sharing their in situ ET observation data of 2015 in
Yanchi Station for our use in calibrating the performance of CLM in this work.

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
