# Peer review of "Determination of appropriate land use/cover pattern based on the mitigating drying effects to support ecohydrological sustainability in the agro-pastoral ecotone of northwest China"

_EGUsphere, 2023_

## Author Response (AR1)

**Reviewer 1**

We would like to thank you for your constructive and thoughtful comments. We have implemented all of your suggestions which have led to a much improved and complete manuscript. In the following sections, the issues raised by you are addressed in the order.

1. The effect of the accuracy of the adopted data on the conclusions should be discussed. For example, accuracy of land use data is 0.1°.

    *Response*: The land use dataset was produced in previous teamwork (Du et al, 2020). 500 samples were selected to evaluate land use data with field investigation and high-resolution images. The evaluation results show that the overall accuracy of the final classification and KAPPA coefficient are both >0.8, which meets the accuracy requirements. The precision of the land use dataset was trustworthy.

    This expression is kept consistent in the paper as follows (L128-L133):

    The surface land use/cover dataset that covered the study area was evaluated in a previous study and the precision was trustworthy (Du et al., 2020). The China meteorology forcing dataset and MODIS LST have been widely used including in the study area of previous work (Li, 2021; Wang et al., 2020). Other datasets like GLASS have been evaluated in the papers that produce the data. The uncertainty of soil properties is in the discussion Section 4.2.

    Du, T., Jiao, J., Duan, H., He, H., XUE, X., and Xie, Y.: Study of conversion between land use/landcover classification system of Chinese Academy of Science and IGBP classification system: In the northwest argo-pastoral zone Journal of Lanzhou University: Natural Science (in Chinese), 56, 91-95, https://doi.org/10.13885/j.issn.0455-2059.20120.01.011, 2020.

    Wang, X., Zhang, B., Xu, X., Tian, J., and He, C.: Regional water-energy cycle response to land use/cover change in the agro-pastoral ecotone, Northwest China, Journal of Hydrology, 580, 124246, https://doi.org/10.1016/j.jhydrol.2019.124246, 2020

    Li, F.: Assessment and fusion of the soil moisture data sets based on community land model and smap satellite (in Chinese), M.S. thesis, Lanzhou Univeristy, 16-40 pp., 2021.

2. Many sentences are very long. It is recommended to shorten them. For example line 45 and line 55.

    *Response*: Thank you. I shortened it and polished all the content in the paper.

    This expression in the paper is as follows (L43-L46):

Satellite products rarely provide accurate continuous long-term data because the satellite obtains instantaneous images, and processing methods introduce uncertainty (Srivastava et al., 2015; Zhang et al., 2010). Numerical models have been used to study multiple variables with high spatial resolution over extended periods and access flux cycles with a consistent framework (Han et al., 2021; Winckler et al., 2018).

3. in line 85. In the form of references.

*Response:* It is a web page. I revised it according to the example of Biogeosciences (shown in the picture).

- Webpages
  - Title
  - URL
  - Access date
  - Year (if not the same as access date)

Copernicus Publications: https://publications.copernicus.org/, last access: 25 October 2018.

This expression in the paper is as follows:

"Additionally, the latest national ecological development project plans to expand grasslands to 60 % in China and continue to convert bare and agricultural lands to grasslands to improve ecosystem services in the APNEC from 2021 to 2035 (China state council, 2017; National development and reform commission, 2019)." (L78)

"The aims of the government plan for 2035 are 1) the grasslands of 60 % and 2) the re-vegetation of bare land and croplands to grasslands (China state council, 2017; National development and reform commission, 2019)." (L309)

In reference:

Major projects for ecological protection and restoration support systems: http://gi.mnr.gov.cn/202006/t20200611_2525741.html, last access: 2024. (L463)

Notice of the state council on printing and distributing the outline of the national land plan (2016-2035): http://www.gov.cn/zhengce/content/2017-02/04/content_5165309.htm, last access: 2024. (L571)

4. The study area in this paper is not the whole of APENC. The specific location of the study area in the APENC should be introduced.

*Response*: The study area in this paper is the agro-pastoral ecotone of Northwest China (APNEWC). It is located in the Northwest of the agro-pastoral ecotone of Northern China (APNEC). So I changed and used APNEWC in ner version.

The boundary of agro-pastoral ecotone are only qualitative descriptions, and the specific distribution boundaries didn't reach an agreement because of the difference of the defined indicators from ecology, climatology, economic geography, and macrogeography. In our project, the agro-pastoral ecotone of Northwest China (APNEWC) was identified by previous research (Wang et al., 2020; Xu et al., 2022; Tan et al., 2020).

This expression is kept consistent in the paper as follows (L93-L97):

"The boundary of the agro-pastoral ecotone did not reach agreement because of the differently defined indicators of ecology, climatology, and economic geography (Li et al., 2021). The APENWC was identified based on previous research (Wang et al., 2020; Tan et al., 2020), including the Otog Banner, Otog Front Banner, Lingwu, Yanchi, Dingbian, Jingbian, Hengshan, Yuyang, Wushen, and Shenmu (Tan et al., 2020; Wang et al., 2021b). It is northwest of the agro-pastoral ecotone of Northern China (APENC)."

Wang, X., Zhang, B., Xu, X., Tian, J., and He, C.: Regional water-energy cycle response to land use/cover change in the agro-pastoral ecotone, Northwest China, Journal of Hydrology, 580, 124246, https://doi.org/10.1016/j.jhydrol.2019.124246, 2020.

Xu, X., Li, X., Wang, X., He, C., Tian, W., Tian, J., and Yang, L.: Estimating daily evapotranspiration in the agricultural-pastoral ecotone in Northwest China: A comparative analysis of the Complementary Relationship, WRF-CLM4.0, and WRF-Noah methods, Sci Total Environ, 729, 138635, https://doi.org/10.1016/j.scitotenv.2020.138635, 2020.

Tan, X., Zhang, L., He, C., Zhu, Y., Han, Z., and Li, X.: Applicability of cosmic-ray neutron sensor for measuring soil moisture at the agricultural-pastoral ecotone in northwest China, Science China Earth Sciences, 63, 1730-1744, https://doi.org/10.1007/s11430-020-9650-2, 2020.

5. The contents of Table 1 can be directly elaborated in paper (delete table 1)

   *Response*: Thank you. I deleted the table.

6. Significance test can be added in Table 5 and Table 6.

   *Response*: Thank you. I added the result of the significant test (Table S3 and Table S4).

Table S3 Relationships between differences in LST and ET and surface albedo, surface roughness, LAI+SAI, aerodynamic resistance, vegetation displacement height, leaf stomatal resistance, and vapor pressure, respectively, in the intense LUCC region ( EXP_grss - EXP_crop). The * indicates statistical significance at the 95% confidence level, and ** indicates statistical significance at the 99% confidence level.

| | $\Delta$ LST | | | | | $\Delta$ Latent heat flux/$\Delta$ ET | | | | |
|---|---|---|---|---|---|---|---|---|---|---|
| | MAM | JJA | SON | DJF | year | MAM | JJA | SON | DJF | year |
| $\Delta$ surface | 0.22 | 0.35 | -0.07 | -0.47 | -0.18 | 0.01 | -0.36 | -0.10 | -0.38 | -0.12 |

| | MAM | JJA | SON | DJF | year | MAM | JJA | SON | DJF | year |
|---|---|---|---|---|---|---|---|---|---|---|
| albedo | ** | ** | ** | ** | ** | | ** | ** | ** | ** |
| Δ surface roughness | 0.13 ** | -0.19 ** | -0.06 ** | -0.07 ** | 0.01 * | -0.27 ** | 0.07 ** | -0.27 ** | 0.21 ** | -0.27 ** |
| Δ LAI+SAI | -0.20 ** | -0.19 ** | -0.15 * | 0.02 | -0.21 ** | 0.33 ** | 0.07 ** | 0.30 ** | 0.24 ** | 0.38 ** |
| Δ aerodynamic resistance | 0.01 | -0.04 ** | -0.05 ** | -0.33 ** | -0.12 ** | -0.03 * | -0.05 ** | -0.05 ** | -0.19 ** | -0.04 ** |
| Δ vegetation height | 0.09 ** | -0.22 ** | -0.10 ** | -0.03 * | -0.04 ** | 0.24 ** | 0.07 ** | -0.21 ** | 0.06 ** | -0.20 ** |
| Δ leaf stomatal resistance | 0.11 ** | 0.19 ** | 0.08 ** | -0.03 * | 0.10 ** | -0.15 ** | -0.11 ** | 0.17 ** | 0.07 ** | -0.03 ** |

Table S4 Relationships between differences in LST and ET and surface albedo, surface roughness, and aerodynamic resistance, respectively, in the intense LUCC region ( EXP_grass - EXP_bare). The ** indicates statistical significance at the 99% confidence level.

| | Δ LST | | | | | Δ Latent heat flux/Δ ET | | | | |
|---|---|---|---|---|---|---|---|---|---|---|
| | MAM | JJA | SON | DJF | year | MAM | JJA | SON | DJF | year |
| Δ surface albedo | 0.01 ** | 0.33 ** | -0.02 ** | -0.30 ** | -0.10 ** | -0.14 ** | -0.51 ** | -0.08 ** | -0.34 ** | -0.18 ** |
| Δ surface roughness | 0.06 ** | -0.22 ** | -0.04 ** | 0.15 ** | -0.05 ** | -0.21 ** | 0.13 ** | -0.27 ** | 0.38 ** | -0.24 ** |
| Δ aerodynamic resistance | -0.06 ** | -0.08 ** | 0.01 ** | -0.22 ** | -0.08 ** | -0.05 ** | 0.07 ** | -0.01 ** | -0.22 ** | -0.01 ** |

7. in line 241. November is missing.

*Response*: Thank you. I have corrected the mistake. This expression in the paper is as follows (L223):

"-0.06 ± 0.14 °C in autumn (SON: September & October & November)."

8. The conclusion can take the form of paragraphs and should be condensed.

*Response*: Thank you. I made the conclusion more condensed. This expression in the paper is as follows (L396-L409):

"This study first simulated and quantified the effects of LUCC using CLM5.0, which was verified based on in-situ observations, in the agro-pastoral ecotone of northwest China. Subsequently, five LUCC scenarios were proposed and assessed to identify the optimal mixture of land use/cover in the study region. The main findings are as follows: First, bare land to grasslands reduced LST while croplands to grasslands increased LST. The bare land to grasslands caused an increase in ET whereas croplands to grasslands caused a

decrease in ET. This led to a spatially averaged cooling surface and increased ET from 2000 to 2015 over the study area. Second, an in-depth analysis of the LUCC pattern from 2000 to 2015 revealed that some grids showed warming or drying, whereas one grid showed both drying and warming. Different mixtures of LUCC could lead to different results for re-vegetation projects, which indicates the complicated synergistic effects of bare land and croplands to grasslands as re-vegetation. Finally, assessing the five proposed LUCC scenarios related to the Chinese government's long-term ecological plan by lowering LST and higher Wc, the proper mixture of LUCC in the APENWC in 2035 is approximately 60 % grasslands, 23 % bare land, and 11 % croplands respectively, which will mitigate the drying and warming surface environment. These findings provide useful information to support land management policy/decision-making in the study region."

**Reviewer 2**

The subject of the study and the development of the technical-scientific part is consistent and provides important land management data for a large area, using validated land use models. Also, the results are integrated with a Chinese government plan for land use restoration, providing useful information for policy development.

We would like to thank you for your constructive and thoughtful comments. We have implemented all of your suggestions which have led to a much improved and complete manuscript. In the following sections, the issues raised by you are addressed in the order.

1. However, there are several things that could be improved; it is a text that in many parts is redundant, for example between L 342 and 346 the same idea is expressed twice without sense, in the same page in L 367 the information presented in table 8 (percentages) is repeated and the SAME information is repeated again in lines 433 and 434.

   *Response*: Thank you.

   L342-346 was polished in the new version (L311-L313) as follows:

   "Thus, in 2035, different mixtures of land use/cover were simulated to pursue the proper mixture of land use/cover. First, we set the percentage of grasslands at 60 % by 2035. Then, the percentage of bare land and croplands, 13 and 30 % respectively, decreases in 2035 to meet the increase in grasslands and is set as the maximum in future scenarios."

   I deleted Table 8.

   L 433-434 was deleted.

2.  The equations 1,2 y 3 are widely used and it´s not necessary to present them.

    *Response*: Thank you. I deleted it.

3.  There is a lot of information (ten tables and fifteen figures!) I suggest reducing this by at least half (ideally less) and leaving some as supplementary material.

    *Response*: Thank you. I left 6 figures and 2 tables. I put others into the Supplementary Material. The change is:

| Orginal version | New verison |
| --- | --- |
| Figure 1 | Figure 1 |
| Figure 2 | Figure S1 |
| Figure 3 | Figure S2 |
| Figure 4 | Figure S3 |
| Figure 5 | Figure S4 |
| Figure 6 | Figure S5 |
| Figure 7 | Figure 2 |
| Figure 8 | Figure 3 |
| Figure 9 | Figure 4 |
| Figure 10 | Figure S6 |
| Figure 11 | Figure S7 |
| Figure 12 | Figure S8 |
| Figure 13 | Figure 5 |
| Figure 14 | Figure 6 |
| Figure 15 | Figure S9 |
| Table 1 | delete |
| Table 2 | Table S1 |
| Table 3 | Table S2 |
| Table 4 | Table 1 |
| Table 5 | Table S3 |
| Table 6 | Table S4 |
| Table 7 | Table S5 |
| Table 8 | delete |
| Table 9 | Table 2 |
| Table 10 | Table S6 |

4.  The results are listed again in the conclusions, it is not necessary to repeat them, also the number 1 (L. 422 and 423) in conclusion Are these study results?.

    *Response*: Thank you. I polished it. This expression in the paper is as follows (L396-L408):

    "This study first simulated and quantified the effects of LUCC using CLM5.0, which was verified based on in-situ observations, in the agro-pastoral ecotone of northwest China. Subsequently, five LUCC scenarios were proposed and assessed to identify the optimal mixture of land use/cover in the study region. The main findings are as follows: First, bare land to grasslands reduced LST while croplands to grasslands increased LST. The bare land to grasslands caused an increase in ET whereas croplands to grasslands caused a decrease in ET. This led to a spatially averaged cooling surface and increased ET from 2000 to 2015 over

the study area. Second, an in-depth analysis of the LUCC pattern from 2000 to 2015 revealed that some grids showed warming or drying, whereas one grid showed both drying and warming. Different mixtures of LUCC could lead to different results for re-vegetation projects, which indicates the complicated synergistic effects of bare land and croplands to grasslands as re-vegetation. Finally, assessing the five proposed LUCC scenarios related to the Chinese government's long-term ecological plan by lowering LST and higher Wc, the proper mixture of LUCC in the APENWC in 2035 is approximately 60 % grasslands, 23 % bare land, and 11 % croplands respectively, which will mitigate the drying and warming surface environment. These findings provide useful information to support land management policy/decision-making in the study region."

5. The use of abbreviations is confusing, bareland, cropland, grassland and "BL", "CL", and "GRS" are used indistinctly.

*Response*: Thank you. I deleted all abbreviations and use "bare land", "croplands", and "grasslands".

6. It´s not necesary to paste the URL of information in the text body (L. 85 and 86) and it´s repeated in L 340 and 341 redundant, the should be in the Reference.

*Response*: It is a web page. I revised it according to the example of Biogeosciences (shown in the picture).

- Webpages
  - Title
  - URL
  - Access date
  - Year (if not the same as access date)

Copernicus Publications: https://publications.copernicus.org/, last access: 25 October 2018.

This expression in the paper is as follows:

"Additionally, the latest national ecological development project plans to expand grasslands to 60 % in China and continue to convert bare and agricultural lands to grasslands to improve ecosystem services in the APNEC from 2021 to 2035 (China state council, 2017; National development and reform commission, 2019)." (L78)

"The aims of the government plan for 2035 are 1) the grasslands of 60 % and 2) the re-vegetation of bare land and croplands to grasslands (China state council, 2017; National development and reform commission, 2019)." (L309)

In reference:

Major projects for ecological protection and restoration support systems: http://gi.mnr.gov.cn/202006/t20200611_2525741.html, last access: 2024. (L463)

Notice of the state council on printing and distributing the outline of the national land plan (2016-2035): http://www.gov.cn/zhengce/content/2017-02/04/content_5165309.htm, last access: 2017. (L570)

7.  L. 100-101 the use of averaged instead of average

*Response*: " I used "annual average" in L98, L99, L247.

8.  The use of water conservation, defined in the water balance Eq 4. (L. 212), implies the use of runoff, it is not explained how this term is approached.

*Response*: All terms are from the output of the model, including runoff, which was evaluated by section 2.4 and previous work (Deng et al, 2022; Li, 2021; Wang et al, 2019). This expression will be kept consistent in the paper as follows (L198-L199):

"the other data are the outputs of CLM5.0, whose performance was validated by Li (2021) and the previous section."

Deng, M., Meng, X., Lu, Y., Shu, L., Li, Z., Zhao, L., Chen, H., Shang, L., Sheng, D., and Ao, X.: Impact of climatic and vegetation dynamic change on runoff over the Three Rivers Source Region based on the Community Land Model, Climate Dynamics, https://doi.org/10.1007/s00382-022-06619-0, 2022.

Li, F.: Assessment and fusion of the soil moisture data sets based on community land model and smap satellite (in Chinese), M.S. thesis, Lanzhou Univeristy, 16-40 pp., 2021.

Wang, H., Xiao, W., Zhao, Y., Wang, Y., Hou, B., Zhou, Y., Yang, H., Zhang, X., and Cui, H.: The Spatiotemporal Variability of Evapotranspiration and Its Response to Climate Change and Land Use/Land Cover Change in the Three Gorges Reservoir, Water, 11, 1739, https://doi.org/10.3390/w11091739, 2019.

9.  In the line 381-382 a part is explained, in the discussion, which should be explained in the methodology

*Response*: Thank you. I deleted it here and put this part into methodology (L160-164) as follows:

"Additionally, two sensitivity experiments were conducted to examine the role of the biogeophysical characteristics of vegetation. The leaf and stem area index (LAI + SAI) of grasslands was replaced by crop in Yanchi_laisai and canopy height in Yanchi_height (Breil et al., 2020). Sensitivity experiments were conducted only at the most representative Yanchi station to save computation time."

10. It is necessary for a english gramatical review, there are errors in the use of verbs e.g. L 100 "...with and annually averaged temperature..." should be "...with an annual average temperature" or L 346 a verb in the past tense is used to talk about 2035. The use of "Here" repeatedly in the wrong contexts. the use of commas needs also another revision, for example in the title.

    *Response*: Thank you.

    I used "annual average" in L98, L99, L247. I deleted all "here" in my paper. I polished all commas in my paper. But I don't see commas in my title. Can you tell me where it is again?

    The new manuscript has been edited to ensure language and grammar accuracy by professional editors at Editage. The editing certificate is as follows:

**editage**

**Editing Certificate**

This document certifies that the manuscript listed below has been edited to ensure language and grammar accuracy and is error free in these aspects. The edit was performed by professional editors at Editage, a brand of Cactus Communications. The author's core research ideas were not altered in any way during the editing process. The quality of the edit has been guaranteed, with the assumption that our suggested changes have been accepted and the text has not been further altered without the knowledge of our editors.

MANUSCRIPT TITLE

**Determination of appropriate land use/cover pattern based on the hydroclimatic regime to support regional ecological management in the agro-pastoral ecotone of northwest China**

AUTHORS
**Yuzuo Zhu**

ISSUED ON
**April 27, 2024**

JOB CODE
**UZZUO_1**

THE EDITAGE
*Promise*
QUALITY GUARANTEE

editage

**Prabh Grewal**
Senior Vice President - Editage

**editage | helping you get published**

Since 2002, Editage has helped over 430,000 authors publish around 1.2 million research papers in scholarly journals across over 1000 disciplines through editorial, translation, transcription, and publication support services. Editage is a brand of Cactus Communications (cactusglobal.com), a science communication and technology company.

**CACTUS.**

GLOBAL :
+1(833) 979-0061 | request@editage.com

CHINA :
400-120-3020或:021-6020-9400 |
fabiao@editage.cn

editage.com | editage.co.kr | editage.jp | editage.cn | editage.com.br | editage.com.tw | editage.de

11. Now a list of particular comments on the text:

L. 33 "violently" it´s out of context this adjective
*Response*: Thank you. I deleted "violently" for better expression (L34).
L. 35 The LUCC is a global mitigation and adaptation strategy in a local context. I suggest adding "adaptation"

*Response*: Thank you. I added it (L36).

L. 66 "Therefore" and "in this context" are redundant

*Response*: Thank you. I deleted it (L63).

L. 77 using an abbreviation such as "W" may not be correct depending on the journal's guidelines.

*Response*: Thank you. I changed it to Wc (L192) and replaced it in all paper.

L. 85-86 the urls should be in the corresponding reference

*Response*: It is the same as 7.

L. 100-101 the use of averaged instead of average

*Response*: Thanks. I used "annual average" in L98, L99, L247.

L. 102 The term vegetation types is used to refer to land use.

*Response*: Thank you. I changed it to land use/cover types (L100).

F. 1 The "line" of the river cuts the DEM Legend

*Response*: Thank you. I revised it (Figure 1. in new version).

[Figure]

F. 3  chart number 20 is missing the metrics

*Response*: Thank you. I revised it (Figure S2. in new version).

[Figure]

L. 221 The use of "severely"?

*Response*: Thank you. I changed it to respectively (L205).

L. 234 "here" it´s wrong used in that way

*Response*: Thank you. I revised it (L216).

F. 7 the legend use "mm" should be C°

*Response*: Thank you. I revised it (Figure 2. in new version).

[Figure]

F. 9 The legend "Mean value" it´s not necessary

*Response*: Thank you. I revised it (Figure 4. in new version).

[Figure]

L. 322 "here" it´s wrong used in that way

*Response*: Thank you. I deleted it (L292).

L 341-342 the urls should be in the corresponding reference

*Response*: It is the same as 7.

L. 346 use of "was" talking about 2035

*Response*: Thank you. I revised it. This expression in the paper is as follows(L312-315):

"Then, the percentage of bare land and croplands, 13 and 30 % respectively, decreases in 2035 to meet the increase in grasslands and is set as the maximum in future scenarios."

L. 348 I dont understand "so five escenarios, which consider computing time, to represent...."

*Response*: We want to propose different scenarios to meet the requirements of the government. This is to say, the grasslands will increase to 60 %, bare land will decrease from 30 %, and croplands will decrease from 13 % during 2015 to 2035. Countless combinations satisfy this requirement. To save computed time, five scenarios were selected to represent the future.

This expression in the paper is as follows(L313-316):

"Subsequently, to reduce computational time, five scenarios were selected to represent the future. The percentage of grasslands, bare land, and croplands were respectively 60, 21, and 13 % in EXP_602113; 60, 23, and 11 % in EXP_602311; 60, 25, and 9 % in EXP_602509; 60, 27, and 7 % in EXP_602707; 60, 30, and 4 % in EXP_603004."

L.349-350 Repeated information with table 8

*Response*: Thank you. I deleted Table 8.

L 359. It is repeated many times that the climatic forcings will be left static to isolate the impacts of LUCC, it is not necessary to repeat so many times that, saying that the climatic forcings remain static the rest is understood.

*Response*: Thank you. I revised it. This expression in the paper is as follows:

"Using static climatic forcings, we compared the difference between future land use/cover scenarios for 2035 and 2015." (L318)

"We ran two experiments in CLM5.0 with two land use/covers (2000 and 2015) and static climatic forcing." (L216)

"Similar to the LST, we only considered the changes in ET directly caused by the LUCC with static climatic forcing." (L223)

L 440. What does it mean that data will be available upon reasonable request? Normally the data is available without conditions.

*Response*: Thank you. I revised it. "The data will be made available on request" (L410).

**Editor**

We would like to thank you for your constructive and thoughtful comments. We have implemented all of your suggestions which have led to a much improved and complete manuscript. In the following sections, the issues raised by you are addressed in the order.

1. A main motivation of this study is to propose land cover mixtures that maintain a sustainable ecohydrological environment (abstract). And the methodological approach is to find an optimal land cover that leads to a cooling surface and higher water conservation. What are the limitations of this approach? Are the temperature and water balance the key variables modulating sustainable ecohydrological environment? What about biodiversity, surface runoff, erosion, etc.? Please discuss about this.

   This expression in the paper is as follows(L363-394):

[revised manuscript text omitted]

3. Revise the Discussion to clarify the main messages of the subsections. E.g., 4.1 Role on what? 4.2 Uncertainties related to soil properties?

*Response*: Thank you. I revised it as follows:

4.1 Sensitivity of LAI + SAI and vegetation height (L331)

4.2 Uncertainty of soil properties (L346 )

4. I agree with reviewer 2 regarding the need to carefully review the entire text and correct the grammatical errors. The grammatical errors in the current version of the manuscript hinders readability and communication of the scientific message.

Thanks! The new manuscript has been edited to ensure language and grammar accuracy by professional editors at Editage. The editing certificate is as follows:

[Figure]

5. Fig. 1: I recommend to change the colors since forests and grasslands are too similar.

   *Response*: Thank you. I revised it (Figure 1. in new version).

---

## Author Response (AR2)

Dear authors,

Thank you for responding to the feedback from both referees. The referees have given significant comments and suggestions that need to be incorporated into a revised version of the manuscript for further review consideration. Alongside addressing the referees' comments, please take into account the following comments for further review consideration.

General comment:
1 The main objective and experimental setup of the study should be better explained in the abstract and introduction sections. For example, what are the experienced negative impacts of the historical land cover changes? And what is you want to mitigate with alternative land cover mixtures?

*Response*: I revised the abstract section: The Chinese government has implemented large-scale vegetation restoration projects and plans to expand the percentage of grasslands to 60 %. However, excessive vegetation restoration consumes more moisture and causes soil drying in the agro-pastoral ecotone of Northwest China (APENWC). The optimal mixture of land use/cover in the APENWC incorporated into vegetation restoration strategies to mitigate the drying effects remains unclear *(The research gap)*. To fill this gap, the Community Land Model version 5.0 (CLM5.0) with static climatic forcing was used to analyse the spatially averaged impacts of land use/cover change (LUCC) by simulating real LUCC scenarios from 2000 to 2015 and examine the impacts of different types of LUCC by simulating idealised maximised LUCC scenarios *(First experimental setup and first objective)*. The results showed…... Furthermore, to identify the proper land use/cover pattern to mitigate drying, we designed different LUCC scenarios by varying the mixture of land use/cover in the CLM5.0 and compared the criteria (water conservation and LST) from the output *(second experimental setup and second objective)*. Based on higher water conservation and cooling surface, results show that ……
*(The results have been omitted here to give a clearer expression of the main objective and experimental setup)*

This study identifies the optimal mixture of land use/cover to expand the percentage of grasslands to 60 % and mitigate the drying effects due to vegetation restoration, for the first time *(The main aim of this paper)*. Previous studies have optimised land use/cover by setting different weights for economic profit and ecological parameters in scenario simulations using a Multi-Objective Genetic Algorithm (Kaim et al., 2018; Kucsicsa et al., 2019; Yang et al., 2020). The experimental design was limited owing to insufficient theoretical studies on parameter settings (Ding et al., 2021) and could not meet the government's preset values (e.g., 60 % grassland) *(research gap)*. Therefore, we first quantified the impacts of the LUCC by simulating land uses/cover scenarios with static climatic forcing in the CLM. We investigated the spatially averaged impacts of LUCC in real scenarios from 2000 to 2015, as well as the impacts of a single type of LUCC in idealised scenarios with maximised LUCC. *(First experimental setup)* These spatially averaged impacts were attributed to the synergy of different LUCC types *(First objective)*. In addition, we further designed different LUCC scenarios by setting the percentage of grasslands as 60 % and varying the percentages of croplands and bare land in the CLM5.0 to identify the optimal mixture of land use/cover in the APENWC based on the higher water conservation and cooling surface. *(second*

*experimental setup and objective).*

There is always a trade-off between the introduction of plants and water consumption (Jia et al., 2017a). Artificial plants consume more moisture, rapidly depleting local soil moisture and leading to a dry layer in the loess profile (Ren et al., 2018; Fu et al., 2017). Soil drying by excessive re-vegetation in the study area has been reported (Jia et al., 2017b; Zhang et al., 2018). Therefore, the negative impact is soil drying. I deleted the word "negative impacts" and used "soil drying" to be more specific. This paper wants to mitigate the drying effects with alternative land cover mixtures by criteria (higher water conservation and lower LST).

This expression is kept consistent in the paper as follows:

[revised manuscript text omitted]

3 The authors make projections for 2035, however, the atmospheric forcing data for the future is not specified in the experimental setup. What are the climatic scenarios you are using to evaluate these projections?

*Response*: The study doesn't deal with future climate scenarios. That's a misrepresentation. My aim is not projections. I revised the whole article about the wrong expression.

Previous studies isolated respectively the effects of LUCC and climate change to better understand the water and energy processes in APENWC (Xue et al., 2019; Wang et al., 2020) and found that vegetation restoration induces soil drying in APENWC (Yang et al., 2021a). This paper aims to mitigate drying effects by adjusting the mixture of different land use/cover in vegetation restoration. It is the research gap we want to fill and why we only consider the impacts of the LUCC. So we keep the atmospheric forcing field constant, thereby isolating the effects of the LUCC. Additionally, to quantify the impacts of LUCC, we ran one control experiment and five sensitive experiments with different land use/covers (shown in Table 1) and static climatic forcing. The differences between the control experiment and sensitive experiments were been used to isolate/quantify the impacts caused by LUCC. This method has been widely used(Breil et al., 2020; Wang et al., 2020). The two results of the impacts of LUCC under climatic forcing in 2000 and 2015 are almost the same (shown in Table 2 and Table 3). This means the impacts of LUCC in our study area are rarely influenced by climate forcing. So we use the atmospheric forcing data of 2015 (Table 3) to represent the result of impacts of different LUCCs and pursue an optimal land use/cover in our study area. I also discuss it in the Discussion section.

Table 1. List of numerical simulations. (It was shown in Table 1 in the manuscript)

**Table 1 numerical simulations for future scenarios.**

| Experiment | Land use/land cover | Forcing | Grid |
|---|---|---|---|
| CN2015 | 2015 | 2000-2015 | 0.1° |
| EXP_602113 | grasslands60%, bare land 21%, croplands13% | 2000-2015 | 0.1° |
| EXP_602311 | grasslands60%, bare land 23%, croplands11% | 2000-2015 | 0.1° |
| EXP_602509 | grasslands60%, bare land 25%, croplands9% | 2000-2015 | 0.1° |
| EXP_602707 | grasslands60%, bare land 27%, croplands7% | 2000-2015 | 0.1° |
| EXP_603004 | grasslands60%, bare land 30%, croplands4% | 2000-2015 | 0.1° |

**Table 2. The spatially weighted averaged differences of LST and WC as different vegetation restoration efforts from 2015 to 2035 under the climate forcing 2000**

|  | $\Delta$ LST (°C) | $\Delta$ WC (mm yr$^{-1}$) |
|---|---|---|
| EXP_602113 | -0.04 | -4.39 |
| EXP_602311 | -0.01 | 0.86 |
| EXP_602509 | 0.02 | 6.09 |
| EXP_602707 | 0.05 | 11.34 |
| EXP_603004 | 0.09 | 19.25 |

**Table 3. The spatially weighted averaged differences of LST and WC as different vegetation restoration efforts from 2015 to 2035 under the climate forcing 2015 (same as the Table 2 in the manuscript)**

|  | $\Delta$ LST (°C) | $\Delta$ WC (mm yr$^{-1}$) |
|---|---|---|
| EXP_602113 | -0.04 | -4.39 |
| EXP_602311 | -0.01 | 0.86 |
| EXP_602509 | 0.02 | 6.09 |

| | | |
|---|---|---|
| EXP_602707 | 0.05 | 11.34 |
| EXP_603004 | 0.09 | 19.25 |

This expression is kept consistent in the paper as follows:

Based on higher water conservation and cooling surface, results show that the optimal percentages of grasslands, bare land, and croplands in the APENWC approximately are 60, 23, and 11 %, respectively, which will mitigate the drying and warming surface environment; this suggests that approximately 5348 km2 of bare land and 1163 km2 of croplands will be transformed into grasslands. (Abstract L26-L29)

This study identifies the optimal mixture of land use/cover to expand the percentage of grasslands to 60 % and mitigate the drying effects due to vegetation restoration, for the first time. (L84-L85)

The government plan aims to 1) expand the percentage of grasslands to 60 % and 2) transform bare land and croplands into grasslands (China state council, 2017; National development and reform commission, 2019). (L311-L312).

Comparing the present land use/cover, EXP_602113 and EXP_602311 resulted in a cooling surface, whereas EXP_602509, EXP_602707, and EXP_603004 resulted in a warming surface. Additionally, EXP_602113 induced drying, whereas EXP_602311, EXP_602509, EXP_602707, and EXP_603004 induced high WC (Table 2) (L320-L322)

Previous studies isolated respectively the effects of LUCC and climate change to better understand the water and energy processes in APENWC (Xue et al., 2019; Wang et al., 2020). Yang et al. (2021a) found that vegetation restoration induces soil drying in APENWC. This paper aims to mitigate drying effects by adjusting the mixture of different land use/cover in vegetation restoration. It is the reason why we only consider the impacts of the LUCC. We designed the experiments with different land use cover/use and static climatic forcing. This method has been widely used to eliminate the influence of other factors and isolate the effects of the LUCC (Wang et al., 2021b; Breil et al., 2020). However, the water and energy processes are affected by changes in both LUCC and climate and vegetation-climate coupling is a complex process. It is worthwhile exploring the contribution of background climate in the future study. (Discussion L385-L392)

**Specific comments:**
4 Abstract:
"large-scale land use/cover change (LUCC)" What do you mean by large-scale?
*Response:* The "large-scale" LUCC is used to differentiate the local-scale LUCC. The former's scope area is larger and time span is longer. There is no clear numerical value of area scope and time span to define the difference between the large-scale LUCC and local-scale LUCC. The obvious signal is that the former influences regional and even global climates through atmospheric circulation (Claussen et al., 2001; Bathiany et al., 2010; Bala et al., 2007; Betts et al., 2007)
In China, the term "large-scale" was used to describe LUCC due to: (1) duration and extent. Since the 1980s, the Chinese government has initiated several large-scale revegetation programs such as "Grain for Green", and "Northern China's Vegetation Belt". These efforts have resulted in a 10%

increase in the leaf area index and an approximately 41.5 million hectares increase in forest area (Li et al., 2018b). Additionally, the Chinese government has planned to expand forests by approximately 220,000 km² (Jia et al., 2017a). (2) interaction with regional and global Climate. For example, the LUCC over eastern China has produced an anomalous cyclonic circulation from the surface to the mid-troposphere over northeastern China and the Korean Peninsula, resulting in increased rainfall (Hu et al., 2015). The LUCC on the Tibetan Plateau intensified The Indian summer monsoon and weakened East China summer monsoon, leading global precipitation slightly increases (Cui et al., 2006). Therefore, we use "large-scale" to describe the vegetation restoration in China.

In APENWC, The croplands decreased by 3922 km$^2$ and the grasslands increased by 6372 km$^2$ from 1993 to 2010 (Wang et al., 2020). The LUCC in the APENWC enhances the moisture recycling process and contributes more precipitable water (Wang et al., 2020; Wang et al., 2021b). Previous studies used the term "large-scale" in APNEWC to describe LUCC (Table 4), so I used "large-scale" LUCC in the study area in my original manuscript. But thanks for your question/suggestion, I think it's not proper and accurate after the literature review since no present study has reported the LUCC in APNEWC will influence the other regions and global climate by atmospheric circulation. Therefore, we deleted "large-scale" to describe the vegetation restoration in APENWC to be more accurate in this paper.

This expression is kept consistent in the paper as follows:

The Chinese government has implemented large-scale vegetation restoration projects and plans to expand the percentage of grasslands to 60 % (**Abstract** L8-L9).

The agricultural pastoral ecotone in Northwest China (APENWC), which is mainly interlaced by grasslands, croplands and bare land, is one of the largest agropastoral ecotones worldwide (Li et al., 2018a; Xue et al., 2019; Yang et al., 2021a). Since the 1980s, The land surface vegetation has been experiencing changes over the last decades due to implemented policies, such as the "Grain for Green Project" and "Three-North Shelterbelt" (Cao et al., 2015; Wei et al., 2018; Liu et al., 2019) (L69-L73).

Table 4 the previous research that uses the term "large-scale" related to the study area

| Research | Term | Study area |
|---|---|---|
| Wang et al. (2020) | Large-scale change in land use/cover | APENWC |
| Jia et al. (2017b) | Large-scale afforestation | The Chinese Loess Plateau (the APENWC is located in the northwest) |
| Xu et al. (2022) | Large-scale ecological restoration projects | APENWC |

Note: Wang and Xu don't use the APENWC as the name of the study area.

5 "Negative environmental effects of excessive re-vegetation have emerged."
What effects?
*Response:* There is always a trade-off between the introduction of plants and water consumption (Jia et al., 2017a). Artificial plants consume more moisture, rapidly depleting local soil moisture and leading to a dry layer in the loess profile (Ren et al., 2018; Fu et al., 2017). Soil drying by

excessive re-vegetation in the study area has been reported (Jia et al., 2017b; Zhang et al., 2018). I deleted all "negative effects" and used "soil drying" to be more specific.

This expression is kept consistent in the paper as follows:
However, excessive vegetation restoration consumes more moisture and causes soil drying in the agro-pastoral ecotone of Northwest China (APENWC). (L9-L10)

6 "Sustainable ecohydrological environment" What is a sustainable ecohydrological environment? Does it mean biodiversity? Water provision? Please specify.
*Response*: I deleted "sustainable ecohydrological environment" and used "ecohydrological sustainability" to keep consistent in my paper. Ecohydrological sustainability studies the interaction between water and ecological systems. This paper focuses on water conservation. The impacts of other factors are limited or have been considered in WC. The reasons are as follows:

Ecohydrological sustainability highlights water as a key driver of ecosystem service (Zalewski, 2021). The ecohydrological sustainability related to water consists of water provision, soil loss, and biodiversity. (1) Water provision. WC is defined as the difference between the income and expenditure of water. It represents the ability of an ecosystem to store or retain water. Therefore, WC represents the amount of water that can be supplied to the region's interior (Bai et al., 2019; Costanza et al., 1997). (2) Soil erosion. Severe soil erosion causes a widespread loss of topsoil and convert the once-flat plateau into hills and gullies, leading to catastrophic floods and droughts on the Loess Plateau of China (Chen et al., 2007; Fu et al., 2017). Since the 1990s, vegetation restoration converted sloping (more than 15°) farmland into forests and grasslands, leading to a soil-retention rate of 84.4% on slopes of 8°-35° (Fu et al., 2017). However, in most areas of APENWC, soil erosion was 0–200 (t $km^{-2}$ $yr^{-1}$) in 2000 and 2008 (Fu et al., 2011), and the soil erosion rate showed no significant change during the Grain-for-Green Project (Fu et al., 2017). This is because APNEC is not a gully-hilly area, where intense soil erosion occurs. The influence caused by soil erosion due to vegetation restoration on the ecohydrological sustainability of APENWC is limited. (3) Biodiversity. Water content between 20 and 60 cm soil depth and soil properties in the study area can be regarded as the primary factors explaining plant and soil fungal diversity (Yang et al., 2017; Wang et al., 2021a). The influence of soil water content on ecohydrological sustainability is included in WC. Additionally, Deng (2022) reported that WC is the crucial factor that needs to be improved in the APENWC based on the ecological sustainability evaluation of vegetation restoration. WC has been used as a type of regulating the ecohydrological sustainability due to LUCC (Deng, 2022; Bai et al., 2019; Zeng and Li, 2019). Therefore, the enhancement of ecohydrological sustainability in the study area mainly focuses on leading to a higher water conversation.

This expression is kept consistent in the paper as follows:
The optimal mixture of land use/cover in the APENWC incorporated into vegetation restoration strategies to mitigate the drying effects remains unclear. (L10-L11)

Ecohydrological sustainability studies the interaction between water and ecological systems, highlighting water as a key driver (Zalewski, 2021). The ecohydrological sustainability related to

water consists of water provision, soil erosion, and biodiversity. (1) Water provision. WC is defined as the difference between the income and expenditure of water. It represents the ability of an ecosystem to store or retain water. Therefore, WC represents the amount of water that can be supplied to the region's interior (Bai et al., 2019; Costanza et al., 1997). (2) Soil erosion. Severe soil erosion causes widespread loss of topsoil and the conversion of the once-flat plateau into hills and gullies, leading to catastrophic floods and droughts in the Loess Plateau of China (Chen et al., 2007; Fu et al., 2017). Since the 1980s, vegetation restoration converted sloping (more than 15°) farmland into forests and grasslands, leading to a soil-retention rate of 84.4 % on slopes of 8 °–35 ° (Fu et al., 2017). However, in most APENWC areas, the soil erosion was 0–200 (t km$^{-2}$ yr$^{-1}$) in 2000 and 2008 (Fu et al., 2011), and the soil erosion rate showed no significant changes during the Grain-for-Green Project (Fu et al., 2017). This is because APNEC is not a gully-hilly area, where intense soil erosion occurs. Therefore, the influence of soil erosion due to vegetation restoration on the ecohydrological sustainability of the APENWC is limited. (3) Biodiversity. During vegetation restoration, the diversity of soil fauna and fungal communities increases, because fast-growing plant species produce large amounts of litter and root exudates, and external resources continually enter the soil food web, promoting nutrient cycling (Wu et al., 2021; Yang et al., 2021b). Water content between 20 and 60 cm soil depths and soil properties can be regarded as the primary factors explaining plant and soil fungal diversity, regardless of land use/cover type (Yang et al., 2017; Wang et al., 2021a). The influence of soil water content on ecohydrological sustainability was included in WC. Additionally, Deng (2022) reported that WC is a crucial factor that needs to be improved in APENWC based on the ecological sustainability evaluation of vegetation restoration. WC has been used as a type of regulating the ecohydrological sustainability due to LUCC (Deng, 2022; Bai et al., 2019; Zeng and Li, 2019). Therefore, the enhancement of ecohydrological sustainability in the study area mainly focuses on improving water conservation. (L364-L384)

7 What do you mean by "hydroclimatic impacts"?
*Response:* Thanks! I deleted the "hydroclimatic impacts" and changed to other expressions.
The water and energy processes refer to the continuous exchange of vapor and heat at the interface between the land and the atmosphere under the drive of atmospheric circulation and solar radiation forcing (Perrier, 1982; Dickinson, 1983). LUCC alters water and energy exchanges between the atmosphere and land surface, leading to changes in ET, LST, and water conservation(Cherubini et al., 2018; Zeng and Li, 2019). I used "hydroclimatic impacts of LUCC" to express "the LUCC's impacts on the water and energy processes" in the original manuscript. I deleted the "hydroclimatic impacts" because "hydroclimatic impacts" is not an academic term. I changed to other expressions such as "water and energy processes response to bare land to grasslands".

This expression is kept consistent in the paper as follows:

Determination of appropriate land use/cover pattern based on the mitigating drying effects to support ecohydrological sustainability in the agro-pastoral ecotone of northwest China (Title)

Furthermore, to identify the proper land use/cover pattern to mitigate drying, we designed

different LUCC scenarios by varying the mixture of land use/cover in the CLM5.0 and compared the criteria (water conservation and LST) from the output. (L24-L26)

Criteria of appropriate land use/cover pattern (Title 2.5 Line 192)

Analyses of the water and energy processes response to bare land to grasslands were conducted in bare land to grasslands and grasslands to bare land intense grid cells (L243-L245)

8 "Our findings suggest the percentages of grasslands, bare land and croplands in the APENWC for 2035 approximately is 60, 23, and 11 %, respectively" It is not clear where these land cover projections come from. Is this your recommendations for an optimal land cover mixture that mitigate the negative impacts? how much of the current land cover should be changed in order to achieve the optimal mixture you are proposing?

*Response:* We designed different LUCC scenarios by setting the percentage of grasslands as 60 % and varying the percentage of croplands and bare land in the CLM and compared the criteria (higher water conservation and cooling surface) from the output. Based on higher water conservation and lowering surface temperature, the optimal mixture of LUCC to mitigate drying was found. We revise the text and tables of experimental design in paper:

**Table 5. List of numerical simulations. (same as the Table 1 in the manuscript)**

| Experiment | Region/points | Land use/land cover | Atmospheric Forcing | Grid |
|---|---|---|---|---|
| Yanchi_grass | Yanchi | grasslands | 2015-2018 | 0.0001 ° |
| Yanchi_crop | Yanchi | croplands | 2015-2018 | 0.0001 ° |
| 18_grass | 18 | grasslands | 2015-2018 | 0.0001 ° |
| 20_grass | 20 | grasslands | 2015-2018 | 0.0001 ° |
| 39_grass | 39 | grasslands | 2015-2018 | 0.0001 ° |
| 42_crop | 42 | croplands | 2015-2018 | 0.0001 ° |
| CN2000 | Domain | 2000 | 2000 | 0.1 ° |
| CN2015 | Domain | 2015 | 2015 | 0.1 ° |
| EXP2000 | Domain | 2000 | 2015 | 0.1 ° |
| EXP2015 | Domain | 2015 | 2000 | 0.1 ° |
| EXP_grass | Domain | Grasslands | 2015 | 0.1 ° |
| EXP_bare | Domain | Bare land | 2015 | 0.1 ° |
| EXP_crop | Domain | Croplands | 2015 | 0.1 ° |
| EXP_602113 | Domain | grasslands60%, bare land 21%, croplands13% | 2000-2015 | 0.1 ° |
| EXP_602311 | Domain | grasslands60%, bare land 23%, croplands11% | 2000-2015 | 0.1 ° |
| EXP_602509 | Domain | grasslands60%, bare land 25%, croplands9% | 2000-2015 | 0.1 ° |
| EXP_602707 | Domain | grasslands60%, bare land 27%, croplands7% | 2000-2015 | 0.1 ° |
| EXP_603004 | Domain | grasslands60%, bare land 30%, croplands4% | 2000-2015 | 0.1 ° |
| Yanchi_laisai | Yanchi | Yanchi | 2015 | 0.0001 ° |
| Yanchi_height | Yanchi | Yanchi | 2015 | 0.0001 ° |

It is the recommendation for an optimal land cover mixture that mitigates drying.

this suggests that approximately 6.9 % (5348 km2) of bare land and 1.5 % (1163 km2) of croplands will be transformed into grasslands to achieve the optimal mixture of LUCC.

This expression is kept consistent in the paper as follows:

Furthermore, to identify the proper land use/cover pattern to mitigate drying, we designed different LUCC scenarios by varying the mixture of land use/cover in the CLM5.0 and compared

the criteria (water conservation and LST) from the output. Based on higher water conservation and cooling surface, results show that the optimal percentages of grasslands, bare land, and croplands in the APENWC approximately are 60, 23, and 11 %, respectively, which will mitigate the drying and warming surface environment; this suggests that approximately 5348 km$^2$ of bare land and 1163 km$^2$ of croplands will be transformed into grasslands. (Abstract L24-L29)

In addition, we further designed different LUCC scenarios by setting the percentage of grasslands as 60 % and varying the percentages of croplands and bare land in the CLM5.0 to identify the optimal mixture of land use/cover in the APENWC based on the higher water conservation and cooling surface. (Introduction L92-95)

To explore a proper mixture of land use/cover in the APENC, we set the percentage of grasslands as 60 % and varied the percentages of croplands and bare land in EXP_602113, EXP_602311, EXP_602509, EXP_602707, and EXP_603004. (L163-L165)

Finally, assessing the five proposed LUCC scenarios related to the Chinese government's long-term ecological plan based on lowering LST and increasing WC, the optimal mixture of LUCC in the APENWC is approximately 60 % grasslands, 23 % bare land, and 11 % croplands respectively; this suggests that approximately 6.9 % (5348 km$^2$) of bare land and 1.5 % (1163 km$^2$) of croplands will be transformed into grasslands to achieve the optimal mixture of LUCC. (Conclusion L405-409)

---

## Author Response (AR3)

Dear authors,

Thank you for responding to the feedback from both referees. The referees have given significant comments and suggestions that need to be incorporated into a revised version of the manuscript for further review consideration. Alongside addressing the referees' comments, please take into account the following comments for further review consideration.

General comment:
1.  Line 143: "SP", full name if it is the first-time usage.
Response: I revised it as 'SP (Satellite Phenology)'.

2. Table 1: The last two sets of experiments seem to be confused, the author should revise the description. And, the temporal resolution of these experiments is not clear, more detailed information is needed here.
Response: I revised the table and context. The model outputs were configured with a temporal resolution of 3 h. (Line 140).

This expression is kept consistent in the paper as follows:

Finally, additional sensitivity simulations were conducted in which specific biogeophysical parameters were altered while other settings remained the same as those in the Yanchi_grass simulation. In the Yanchi_laisai simulation, the leaf and stem area index (LAI + SAI) of grasslands was replaced by LAI + SAI of cropland. Similarly, in the Yanchi_height simulation, the vegetation height of grasslands was substituted with the vegetation height of cropland (Breil et al., 2020). These sensitivity experiments aimed to explore the influence of biogeophysical factors in regulating energy and vapour fluxes during LUCC. (Line 154-158).

3. Why did you design the EXP2015? No analysis of this experiment was found in the manuscript.
Response: Thanks, I deleted it.

4. Line 315: The authors should provide more detailed descriptions about why the sum percentage of bare land and croplands is 34%, which is different from Line 314.
Response: These experiments assumed that land use conversions occur exclusively among barren land (29.9%), croplands (12.5%), and grasslands (52.0%), while other land use/cover classes (5.6%) were excluded from the transformation process. So, in the experiments, we set other land use/cover classes (~ 6%) not to change. The grassland is 60%, so the sum of bare land and croplands is 34%.

This expression is kept consistent in the paper as follows:

*These experiments assumed that land use conversions occur exclusively among barren land (29.9%), croplands (12.5%), and grasslands (52.0%). While other land use/cover classes (5.6%), including shrublands, urban areas, water bodies, and forests, remained constant and were excluded from the transformation process. Two extreme scenarios were defined: in EXP_603004,*

*additional grassland is converted entirely from croplands, resulting in a land use/cover composition of 60% grassland, 30% bare land, and 4% croplands; in EXP_602113, grassland expansion occurs through the conversion of bare land, resulting in a composition of 60% grassland, 21% bare land, and 13% croplands. Additionally, three intermediate scenarios were designed with incremental variations in the proportions of bare land and croplands: 60%, 23%, and 11% in EXP_602311; 60%, 25%, and 9% in EXP_602509; 60%, 27%, and 7% in EXP_602707. (Line 288-295)*

5. Line 318: As shown in Table 1, these experiments were conducted under the dynamic climatic forcing, not "static climatic forcing".

*Response*:  I delete 'static climatic forcing'. We conducted experiments under the dynamic climatic forcing. I review previous studies with similar simulations. A couple study expressed as express as 'assuming fixed oceanic conditions'(Davin and De Noblet-Ducoudré, 2010) or 'keeping the atmospheric forcing field constant' (Wang et al., 2020). While some studies only explain LUCC experimental design with different land use/cover and the same simulation period. They don't emphasize that the climatic forcing doesn't change (Breil et al., 2020; Cherubini et al., 2018). I revised the experiment in the paper as follows:

*A suite of land use/cover scenarios experiments were designed to explore the impacts of LUCC. The only difference among the land use/cover scenario experiments was the land use/cover class, ensuring that the impacts of LUCC were isolated. The impacts of LUCC from 2000 to 2015 were quantified by comparing EXP2000 and CN2015 (Wang et al., 2020; Breil et al., 2020). Three additional experiments examined the effect of individual LUCC: EXP_bare and EXP_crop scenarios respectively extended bare land and croplands to 100%, respectively, while EXP_grass set grasslands to 100%, replacing bare land and croplands (Cherubini et al., 2018). To evaluate vegetation restoration in the APENWC, future land use/cover scenarios were conducted by setting the percentage of grasslands at 60% and varying proportions of croplands and bare land simulated in EXP_602113, EXP_602311, EXP_602509, EXP_602707, and EXP_603004 (Line 146-153)*

General comment:
Why the optimal scenario was selected based on the minimum delta WC? (Table 2). Would an increase in WC be desirable in terms of sustainability? There are other scenarios that decrease ET while keeping the 60% target in grassland. A scenario of "minimum change" is not necessarily an optimal scenario. It depends on what is the main objective of the restoration project and the reference conditions these variations are computed from.

There are scenarios showing a slight warming (higher LST) and increases in water balance (higher WC), based on the arguments provided in the manuscript (copied below), the scenarios with higher WC should be selected. Please clarify:

"For sustainable ecological construction, it is necessary to pursue an alternative mixture of land use/cover without augmenting warming and endangering water availability. This means the proper mixture of land use/cover has a lower LST and larger WC than in 2015 (Arora and Montenegro,

2011; Bai et al., 2019; Wang et al., 2021d; Findell et al., 2017). Therefore, vegetation restoration strategies in the APENWC should use an appropriate mixture of land use/cover, such as EXP_602311; this indicates that approximately 6.9 % (5348 km2) of bare land and 1.5 % (1163 km2) of croplands transformed into grasslands. The LUCC to EXP_602311 generally caused more cooling and slightly increased the WC owing to proper vegetation restoration. Otherwise, other scenarios would lead to warming or drying, exacerbating drought in the APENWC."

Response: It's true that it's not proper to determine the optimal scenario. I will change my abstract/result/conclusion to be more academic.

Excessive vegetation restoration undermined soil drying in the APENWC. In this paper, I design future land use/cover scenarios by setting grassland coverage at 60% and varying the proportions of croplands and bare land in CLM5.0. These scenarios were evaluated to assess the potential impacts of future LUCC in the APENWC, emphasizing ecohydrological sustainability/water conservation. The results show that none of the scenarios showed significant adverse effects on WC, suggesting that vegetation restoration will not intensify drying conditions. These results indicate that increasing grassland coverage to 60% by 2035 supports ecohydrological sustainability without introducing drying.

This expression is kept consistent in the paper as follows:

Title: Expanding Grassland Coverage Maintain Ecohydrological Sustainability in the Agro-Pastoral Ecotone of Northwest China

Abstract:
*To achieve ecological sustainability, the Chinese government is conducting large-scale vegetation restoration projects to increase grasslands to 60 % by 2035. However, excessive vegetation restoration undermined soil drying in the agro-pastoral ecotone of Northwest China ...... It is unclear the potential impacts of future land use/cover change (LUCC) on ecohydrological sustainability over the APENWC. To fill this gap ......* (The result:) *Future scenarios assuming 60 % grassland cover with varying proportions of bare land and cropland suggest that none of the scenarios showed significant adverse effects on WC, suggesting that vegetation restoration will not intensify drying conditions. These results indicate that increasing grassland coverage to 60% by 2035 supports ecohydrological sustainability without introducing drying.*

*Introduction:*
*(The research gap:) Excessive vegetation restoration has been reported to undermine ecohydrological sustainability, leading to challenges such as soil drying (Jia et al., 2017b; Zhang et al., 2018). These findings highlight the urgent need for land use/cover configurations that balance vegetation restoration with ecohydrological sustainability in the APENWC. Additionally, the latest national ecological development plan, implemented from 2021 to 2035, aims to increase grassland coverage to 60 % and convert bare land and croplands into grasslands to enhance ecosystem services ((China state council, 2017; National developmen tand reform commission, 2019). However, this ambitious target has not been thoroughly assessed for its potential impacts on ecohydrological sustainability.*

*……*

*(What I do:)* **This study examined the impacts of historical and future LUCC associated with vegetation restoration projects in APENWC using the Community Land Model (version 5.0, CLM5.0). Section 2 …… Section 3.1 aanalysed the spatially averaged impacts of LUCC during the historical period of 2000–2015 under a realistic LUCC scenario. Additionally, the individual impacts of different LUCC types were quantified using idealised scenarios where specific LUCC types are maximised. Section 3.2.1 categorized the historical land use/cover composition from 2000 to 2015 and attributed the spatially averaged impacts to the synergistic effects of multiple LUCC types. Finally, Section 3.2.2 introduced future LUCC scenarios designed to achieve the government's target of 60 % grassland coverage. These scenarios were evaluated to assess their potential effects on water conservation (WC) in the APENWC. Section 4 …… Section 5 ……**

*Result:*

*The simulations of future land use/cover scenarios, compared with 2015 are presented in Table 2. Grassland expansion through the conversion of croplands (EXP_603004) resulted in significant variations in LST (0.09 ℃) and ET (-17.62 mm yr-1). Achieving the target of increasing grassland coverage to 60 % by 2035, primarily through cropland-to-grassland conversions, would increase ET in the futur. However, none of the scenarios had a significant negative impact on WC, indicating that the vegetation restoration efforts are unlikely to exacerbate drying. These findings suggest that increasing grassland coverage to 60% by 2035 maintains ecohydrological stability while advancing vegetation restoration goals.*

General writing of the manuscript: The response letter and revised manuscript have helped to better understand the methodological framework, results and conclusions; however, the English writing still requires an in-depth revision as some ideas are still unclear or misleading.

Response: Thanks. I revised the abstract following your suggestion and revised all the manuscripts by myself. Then the new manuscript has been edited to ensure language and grammar accuracy by professional editors at Editage. The editing certificate is as follows:

[Figure]

Minor comment: Revise grammatical error: APNEC appears several times. It should be APENWC.

Response: Thanks. I revised it.

Reference

Breil, M., Rechid, D., Davin, E. L., de Noblet-Ducoudré, N., Katragkou, E., Cardoso, R. M.,

Hoffmann, P., Jach, L. L., Soares, P. M. M., Sofiadis, G., Strada, S., Strandberg, G., Tölle, M. H., and Warrach-Sagi, K.: The Opposing Effects of Reforestation and Afforestation on the Diurnal Temperature Cycle at the Surface and in the Lowest Atmospheric Model Level in the European Summer, Journal of Climate, 33, 9159-9179, https://doi.org/10.1175/jcli-d-19-0624.1, 2020.

Cherubini, F., Huang, B., Hu, X., Tölle, M. H., and Strømman, A. H.: Quantifying the climate response to extreme land cover changes in Europe with a regional model, Environmental Research Letters, 13, 074002, https://doi.org/10.1088/1748-9326/aac794, 2018.

Davin, E. L. and de Noblet-Ducoudré, D. N.: Climatic Impact of Global-Scale Deforestation: Radiative versus Nonradiative Processes, Journal of Climate, 23, 97-112, https://doi.org/10.1175/2009jcli3102.1, 2010.

Wang, X., Zhang, B., Xu, X., Tian, J., and He, C.: Regional water-energy cycle response to land use/cover change in the agro-pastoral ecotone, Northwest China, Journal of Hydrology, 580, 124246, https://doi.org/10.1016/j.jhydrol.2019.124246, 2020.